# "Blooming" of litter-mixing effects: The role of flower and leaf litter interactions on decomposition in terrestrial and aquatic ecosystems

Mery Ingrid Guimarães de Alencar[1,2], Rafael D. Guariento[3], Bertrand Guenet[2], Luciana S. Carneiro[1], Eduardo L. Voigt[4] & Adriano Caliman[1]

[1] Departamento de Ecologia, Centro de Biociências, Universidade Federal do Rio Grande do Norte, Natal, 59078-900, Brazil.

[2] Laboratoire de Géologie, Ecole normale supérieure, CNRS, IPSL, Université PSL, Paris, 75005, France

[3] Universidade Federal do Mato Grosso do Sul, CCBS, Campo Grande, 79070-900, Brazil.

[4] Departamento de Biologia Celular e Genética, Centro de Biociências, Universidade Federal do Rio Grande do Norte, Natal, 59078-900, Brazil.

*Correspondence to*: Mery Ingrid Guimarães de Alencar (alencarmery@gmail.com)

**Abstract.** The diversity effect on decomposition, through the litter-mixing effects, plays a central role in determining the nutrient and carbon dynamics in ecosystems. However, the litter-mixing effects are centered on a leaf litter perspective. Important aspects related to intraspecific interaction and biomass concentration are rarely evaluated, even though they could be essential to determine the litter decomposition dynamics. To our knowledge, we introduced a new perspective to evaluate whether and how the interaction between flower and leaf litter affects the occurrence, direction, and magnitude of litter-mixing effects in terrestrial and aquatic ecosystems. We performed laboratory experiments using flower and leaf litter from the yellow trumpet tree *Tabebuia aurea* (Silva Manso) Benth. & Hook. f. ex. S. Moore as a model. To obtain realistic results, we manipulated various scenarios of flower:leaf litter biomass proportion and measured 13 functional traits, respectively. Litter-mixing effects were consistent in both aquatic and terrestrial environments, with faster decomposition of both litter types in mixtures compared to their monocultures (synergistic effects). Litter-mixing effects were stronger in the terrestrial environment and at higher flower:leaf litter biomass proportions. Our results indicate that synergistic outcomes are mainly associated with complementary effects. Flower litter had a higher concentration of labile C-compounds, N, P, and K and lower lignin concentrations representing a labile litter; while leaf litter had a higher concentration of lignin, Ca, Mg, and Na, representing a refractory litter. Our results demonstrate the importance of litter-mixing effects between flower and leaf litter via complementary effects. These results shed light on the secondary consequences of flower litter on decomposition, suggesting that species with high reproductive investment in flower biomass may play an important role in the nutrient and carbon recycling of diverse plant communities, exerting a pivotal role in biogeochemical dynamics.

## 1 Introduction

Decomposition is an important ecosystem process because of its role in the energy and matter flows within and across terrestrial and aquatic ecosystems, which affects the cycling of nutrients and carbon (C) in the biosphere (Cebrian and Lartigue, 2004; Tiegs et al., 2019). Up to 90% of the primary production accumulates as organic matter (OM) in the soil (Cebrian, 1999), and a considerable proportion of this stock is transported to rivers, lakes, and oceans, contributing to the stocks of OM in aquatic environments, along with autochthonous OM (Tranvik et al., 2009; Aufdenkampe et al., 2011). Decomposition is controlled by abiotic factors such as temperature and humidity, as well as biotic factors such as the abundance and composition of decomposers, and litter quality, with the relative importance of each factor varying between biomes and ecosystems (Makkonen et al., 2012; García-Palacios et al., 2016; Djukic et al., 2018). While global OM stocks are relatively well known (Schimel et al., 2001; Hengl

et al., 2017), the mechanisms governing OM dynamics in ecosystems are far less understood (Tian et al., 2015). Given the vast size of OM stocks in ecosystems, even minor changes in OM content and dynamics can have major impacts on global C and nutrient budgets (Dignac et al., 2017; Minasny et al., 2017). Therefore, understanding the peculiarities of the mechanisms that regulate decomposition dynamics is crucial for comprehending the flows and stocks of C and nutrients in terrestrial and aquatic ecosystems (Basile-Doelsch et al., 2015; Davidson and Janssens, 2006).

Litter quality, which refers to the edibility of litter as a food resource for decomposers, has effects on decomposition that are comparable to, or even stronger than, those of abiotic factors across terrestrial and aquatic ecosystems (Makkonen et al., 2012; García-Palacios et al., 2016). The chemical compounds and physical structures of plant litter in ecosystems are highly diverse and heterogeneous, leading to distinct litter quality (Freschet et al., 2010, 2013; Olson and Pittermann, 2019; Schmitt and Perfecto, 2020). The diversity of plant life forms, organs, and traits corresponds to the myriad of plant tissues that contain different pools of compounds (Jackson et al., 2013). After senescence, these tissues form litter pools with a wide range of resistance to biological (mostly microbial) degradation (Kuzyakov and Blagodatskaya, 2015; Jones et al., 2023). Therefore, the variety of functional differences in plant litter can affect decomposition both intrinsically and through mechanisms mediated by complex interactive effects among contrasting litter types, resulting in litter-mixing effects (LMEs) on decomposition (Schindler and Gessner, 2009; Liu et al., 2020; Hättenschwiler and Jørgensen, 2010).

Most biogeochemical models predicting decomposition dynamics in natural ecosystems ignore interactions among plant litters of different qualities and assume that the sum of the individual litters can predict decomposition in a mixture (Manzoni and Porporato, 2009). However, LMEs on decomposition could either increase (i.e. synergistic) or decrease (i.e. antagonistic) the decomposition rate of individual litter compared to their monocultures, and the magnitude of such effects can be highly variable depending on litter traits, decomposer community, and environmental contexts (Gartner and Cardon, 2004; Liu et al., 2020; Porre et al., 2020). The synergistic effects are mainly attributed to mechanisms such as nutrient transfer, niche partitioning among decomposers, and improvement of microclimatic conditions (Gessner et al., 2010; Boyero et al., 2011; Hättenschwiler et al., 2005; Makkonen et al., 2013). Mechanisms responsible for antagonistic effects include microbial nutrient immobilization, inhibitory decomposition by secondary compounds, and decomposer preferential feeding (Kuzyakov, 2002; Hättenschwiler et al., 2005). However, these mechanisms have been predominantly demonstrated in studies that manipulate inter-specific leaf litter diversity and in experiments where species contribute equally to the litter biomass in the mixture (but see Madritch and Hunter 2004; Crutsinger et al. 2009; Zhang et al. 2022). Some efforts have been directed towards understanding the intra-specific LMEs from different plant organs or varying proportions of litter types in litter mixtures on ecosystem functions (Dearden et al., 2006; de Paz et al., 2018; Schmitt and Perfecto, 2020; Hou and Lü, 2021; Zhao et al., 2019; Zhang et al., 2022). This is an important research avenue because it reinforces the idea that LMEs on decomposition depend more on functional dissimilarity than on the taxonomic richness of litter, and it may also indicate that within-species LMEs may occur and be particularly relevant for ecosystem functioning in low-diversity plant communities.

Previous studies have shown that the magnitude of LMEs on decomposition is affected by dissimilarity in litter quality (Schindler and Gessner, 2009; Finerty et al., 2016). As a result, much of the research on LMEs has focused on mixing leaf litter from different species (Porre et al., 2020; Hättenschwiler et al., 2005), whereas the interaction among mixed litters from different plant organs has not been well explored. It is important to note that unequal investment among plant organs, which can occur due to differences in organ form and function, may result in variations in the chemical composition of tissues across different plant organs, which has legacy consequences on litter decomposition (Freschet et al., 2013; Jackson et al., 2013). Leaves are specialized organs for photosynthesis and have a longer lifespan than flowers (Roddy et al., 2019). Therefore, accordingly to the growth-rate hypothesis, leaves are expected to have higher concentrations of structural compounds such as lignin and secondary metabolites

than flowers (Stamp, 2003). Conversely, flowers are fast-growing ephemeral organs specialized in reproduction (Ashman and Schoen, 1994). As a result, flowers, on average, are expected to receive less investment in the production of constitutive defenses against herbivory and structural tissues, but a greater investment in nutrients for growth and in labile C-compounds such as sugars for nectar production to attract pollinators (Mccall and Irwin, 2007; Boaventura et al., 2022). Finally, the differences in litter quality between leaves and flowers could also become more pronounced during senescence, as leaves have been shown to have the highest rate of nutrient resorption among plant organs (Freschet et al., 2010). Thus, since litter characteristics in general reflect environmental pressures that shape the form and function of plant organs when alive, we rationalized that flower litter is more labile than its leaf litter, and that this dissimilarity may cause LMEs in their decomposition when both litters are mixed.

Studies have demonstrated that the mixture of various sources of dissolved and particulate labile organic matter (LOM) and refractory organic matter (ROM) can exert opposing effects on each other's decomposition, both in terrestrial and aquatic ecosystems (Guenet et al., 2010). Generally, LOM is expected to accelerate the decomposition of ROM (Guenet et al., 2010; Wang et al., 2022) while ROM is expected to inhibit the decomposition of LOM (Liu et al. 2020; but see Swan and Palmer 2006; Cuchietti et al. 2014). However, it is currently unclear how variations in the relative proportions of LOM and ROM in litter mixtures affect the magnitude and direction of LMEs, considering the degradation rate of each litter type and the whole litter mixture. This is important because the LOM:ROM biomass ratio in the detritus pool varies spatially and temporally within and among ecosystems due to a variety of causes (McClain et al., 2003). The LOM:ROM biomass ratio in the detritus pool is critical for microbial degradation rates, as the limited number of metabolic pathways available to microbial decomposers have specific energy requirements (German et al., 2011). Therefore, the LOM:ROM proportion in litter mixtures could be pivotal in determining the occurrence, magnitude, and direction of LMEs on decomposition (Smith and Bradford, 2003; Schindler and Gessner, 2009), but this is not well understood (Sayer et al., 2007; Gripp et al., 2018).

In this study, we explored a novel potential after-life role of flower litter in mediating the LMEs on decomposition. For this, we utilized flower and leaf litter (hereafter litter types) from the trumpet tree *Tabebuia aurea* (Silva Manso) Benth. & Hook. f. ex. S. Moore, as sources of LOM and ROM, respectively. To better understand the possible mechanisms underlying LMEs on decomposition, we evaluated the occurrence, magnitude, and direction of LMEs on each litter type individually and on whole litter mixture. In doing so, we sought to determine whether the LMEs of flower and leaf litter mixing were reciprocal or unilateral, and whether the magnitude and direction (i.e. synergistic or antagonistic) of such effects were symmetric or asymmetric. We tested the following hypotheses: (*i*) flower litter quality will be higher (i.e. LOM) than leaf litter quality (i.e. ROM) and consequently flower litter will decompose faster than leaf litter; (*ii*) the interaction between flower and leaf litter during decomposition will result in LMEs; (*iii*) litter-mixing effects on each litter type will be mostly asymmetric, with more frequent and/or stronger positive effects of flower litter on the decomposition of leaf litter; and (*iv*) litter-mixing effects, on each litter type and mixture, will depend on the relative proportion of each litter type in the mixture. Since the species from the Tabebuia genus as well as other trumpet trees can colonize floodplains, riparian areas, and seasonally dry forests in the tropics (Ribeiro and Brown 2006), the litter of these species can be decomposed under aquatic or terrestrial ecosystem contexts. Litter-mixing effects on decomposition have been traditionally investigated in aquatic (Boyero et al., 2021) and terrestrial (Makkonen et al., 2012) ecosystems separately (but see Handa et al. 2014; García-Palacios et al. 2016) using different species and methodologies. This hinders testing the generality of the results of these studies for different types of ecosystems. Therefore, considering that *T. aurea* can contribute litter to both terrestrial and aquatic ecosystems, we tested our hypotheses throughout terrestrial and aquatic experiments.

.

## 2 Methods

### 2.1 Study site and species

The experiment was conducted in the laboratory at the Universidade Federal do Rio Grande do Norte, Brazil. The flower and leaf litter of *T. aurea* was sampled in a forest fragment (more details below). The geographic distribution of *T. aurea* in South America extends to most Brazilian biomes, such as the Amazon, Atlantic Forest, Cerrado, and Pantanal (Lorenzi, 1992), and its environmental distribution ranges from dry forests to riparian forests and floodplains (Batalha and Mantovani, 2001; Lorenzi, 1992). Thus, the widespread geographic and environmental distribution of *T. aurea* allows its litter to contribute to the flow of matter and energy in both the aquatic and terrestrial ecosystems (Fig. S1; see Supplement A for more details to species in the section *Species used*). Synchronous and massive flowering, which is a common characteristic of Bignoniaceae species, is preceded by the loss of leaves (Barros, 2001). This phenological pattern creates a potentially important scenario for testing the LMEs on decomposition, as a layer of leaf litter is deposited in the soil, which is then covered by a layer of flower litter a few days later (Fig. S1).

**2.2 Estimation of flower and leaf litter functional traits**

We measured a set of traits to describe the functional differences between the litter types of *T. aurea*. These analyses focused on estimating the initial values for litter chemical and physical traits that commonly have an impact on litter decomposition in terrestrial and aquatic ecosystems. Each functional trait had three replicates. For chemical analyses, at least 3 g of each litter type was ground to a fine powder using a mortar and pestle. We then estimated the total C concentration using the high-temperature combustion method and infrared $CO_2$ detection with a Shimadzu TOC-5000 total carbon analyzer. Total nitrogen content (N) was estimated by acid digestion using Kjeldahl distillation (Allen et al. 1974). The total phosphorus (P) was estimated through strong acid digestion and reaction with molybdate (Fassbender, 1973). Potassium (K), calcium (Ca), and manganese (Mg) were determined in flame atomic emission spectroscopy after nitro-perchloric digestion (Sarruge and Haag, 1974). Sodium (Na) content was estimated via flame atomic emission spectroscopy (Robertson et al., 1999). Structural compounds, such as lignin (Lig) and cellulose (Cel), were estimated by the sequential method of neutral detergent and second acid detergent digestion (Goering and Van Soest, 1970). Phenolic compounds (Phe) were estimated by the Folin assay (Graça et al., 2005). We used the Antrona method (Morris, 1948; Van Handel, 1968) to determine the non-reducing soluble sugars (S-carb).

To evaluate physical traits, we assessed the water-holding capacity (WHC) and leaching of flower and leaf litter, which are considered crucial factors in determining litter decomposition (Makkonen et al., 2013). To evaluate WHC, we used dried flower and leaf litter and moistened the replicates with 50 ml of water (the same volume used to irrigate the terrestrial experiment) two hours before the measurements, based on Makkonen et al. (2013). For the leaching measurement, we stimulated the loss of hydrolysable water compounds, which is the main form of mass loss in the initial stage of decomposition, based on Pérez-Harguindeguy et al. (2013). For both structural traits, the material was dried in an oven at 60 °C for 72 h before and after the measurements.

**2.3 Litter sampling and experimental design and setup**

We sampled flower and leaf litter under the canopy of *T. aurea* individuals immediately after abscission. Soon after litter sampling, the litter types were separately dried at 60 °C for 72 h until a constant weight was achieved. The litter was conditioned in a dry and dark place to avoid changes in its chemical composition.

The experimental design followed an additive rather than a substitutive design, which is commonly used in experiments designed to test the effects of species diversity and litter mixing on ecosystem functioning (Jolliffe, 2000) (Table S1; see Supplement A for more detailed description about the additive experimental design in *Experimental design and setup*).

We performed controlled laboratory experiments to simulate terrestrial and aquatic environments. The duration of the terrestrial and aquatic experiments was standardized by the time required for approximately 50% of the more labile litter (i.e. flowers) to be decomposed in each environment. The aquatic experiment lasted for 3 months, while the terrestrial experiment lasted for 7 months.

To ensure aerobic conditions in both environments, distinct microcosms were used. In the terrestrial experiment, plastic containers (5 cm in diameter and 10 cm in height) were used as microcosms. Each terrestrial microcosm was filled with a soil layer of approximately 5 cm height, collected under the canopy of *T. aurea* individuals in the same area in which litter was collected. The soil was sieved (2-mm mesh size) to remove large litter particles, homogenized, and added to the microcosms. This procedure maintained a substantial part of the soil microflora and micro- and meso-fauna (Swift et al., 1979) while reducing environmental heterogeneity among experimental microcosms. It is important to note that the flower:leaf biomass proportion in the litter layer can significantly vary in nature across space and time. This variation can be attributed to several factors such as plant species identity, individual size, timing and magnitude of flowering phenology, and distance from the plant originating the litter (Uriarte et al. 2015; Buonaiuto and Wolkovich, 2021). However, despite the significance of this information, the literature still lacks data on flower:leaf biomass proportion for the majority of species, including the species used in our study. Although, a recent study looked at the amount of flower and leaf litter biomass for several species. The study found that despite leaf litter is generally more common than flower litter on an annual basis, the amount of flower and leaf litter varies significantly throughout the year. As a result, the proportion of flower:leaf biomass in the litter layer can vary greatly for different species. On average, flower litter contributes around 25% of the leaf litter on an annual basis, but this can range from 5% to 45% (Hill et al. 2022). And in some cases, during the blooming season, the amount of flower litter can even exceed the amount of leaf litter (Wang et al. 2016). Therefore, to encompass the unknown and possibly extensive variability in flower:leaf biomass proportion that may occur for *T. aurea* in nature, we assembled mixtures of flower and leaf litter along a gradient encompassing nine different flower:leaf biomass proportion. The amounts of litter types were added according to the information in Table S1.

Because *T. aurea* displays a very marked phenological pattern of leaf and flower abscission (where leaves senesce and fall completely one to two weeks before flowering and the consequent flower fall), and because studies have demonstrated that litter spatial position can alter litter decomposition (Berenstecher et al., 2021), we arranged the flower and leaf litter in the microcosms resembling their natural position in the litter layer. First, the leaf litter was added above the soil within the microcosms, followed by the flower litter overtopping it. All microcosms were then randomly arranged in plastic trays and covered with a 1 mm mesh opening screen to prevent the entry of foreign materials. The experiment was conducted in a laboratory room at a constant temperature of approximately 25 °C and a 12:12 h light:dark period. To avoid moisture limitation of litter decomposition in the terrestrial experiment, each microcosm was individually irrigated every three days with approximately 50 ml of tap water using a hand-held sprinkler. The amount of water was based on an estimation of the accumulated average precipitation at the study site during the experiment (January to July; Santos e Silva et al. 2012)

In the aquatic experiment, the microcosms were composed of 1 L glass bottles. The amount of each litter type added to the respective monocultures and mixtures is shown in Table S1. Dechlorinated tap water was used to fill the aquatic experimental microcosms and the water inoculum from the oligotrophic Carcará Lake (6°3´40"S, 35°9´28"W) was added to allow the colonization of microorganisms. The microcosms of the mixtures were filled with 1 L of water. However, as the final litter biomass added to the microcosms differed between mixtures and their respective monocultures, as well as throughout the monocultures, the volume of water in each microcosm was adjusted to maintain a final litter concentration of 3 g/L across all treatments.

As in the terrestrial experiment, litter types were added intact to the microcosms. To prevent litter from floating and/or sticking to the inner wall of the microcosms, we packed litter in synthetic bags with a 1 mm mesh size, each containing a small

metal weight, to keep the litter near the bottom of the microcosms. We ensured aerobic conditions in each microcosm and promoted
water circulation and constant homogenization of abiotic conditions by providing aeration to each individual microcosm. The
microcosms were randomly distributed along shelves in a darkened room at a constant temperature of approximately 25 °C to
avoid primary production.

**2.4 Measurements of the litter mass remaining**

204       At the end of the experiments, litter was carefully removed from the microcosms. For mixtures, the remaining flower

and leaf litter were visually identified and separated and subsequently dried at 60 °C for 72 h and weighed to estimate litter mass loss.
The procedure varied between terrestrial and aquatic experiments. In the terrestrial experiment, we separated the remaining flower
and leaf litter from each other (in mixtures) and from the soil particles and placed them in aluminum trays for subsequent weighing.
In the aquatic experiment, flower litter fragmentation limited a similar procedure. Instead, we filtered the litter from each
microcosm, for both monocultures and mixtures, using a previously weighted paper filter. We used paper filters to quantify the
small particulate organic matter associated with flower litter, which fragmented more easily (personal observation). This problem
did not occur with leaf litter, which disintegrated into larger particles at the end of the experiment. However, to maintain the same
weighing procedure between the two litter types, we filtered them through paper filters and then quantified their mass loss
separately. For monocultures, we removed the flower and leaf litter from the bags and placed them on a paper filter at the end of
the experiment. We then poured the entire water volume from each microcosm containing the leaked particulate material into the
corresponding filter. We followed the same procedure for the mixtures; however, leaf litter fragments, which were tougher than
flower litter, were easily identified and collected from the filter surface. Leaf litter fragments were placed in previously weighed
paper filter. The identified litter on the paper filter was then packed in aluminum trays and dried at 60 °C for 72 h. Subsequently,
we repeated the weighing procedure and measured the remaining mass of each litter type in its respective microcosm, allowing us
to estimate the decomposition rate of each litter type individually, even in mixtures. To verify if the paper filter could retain fine
particles only for flower litter, we compared the paper filter mass before and after filtration of leaf litter to guarantee that there was
no overestimation of flower litter mass due to fine leaf litter particles retained in paper filter, though the comparison of paper filter
mass before and after leaf litter filtration of leaf litter in monoculture treatments (t-test= 0.95; p=0.78). We estimated the
decomposition rate in both experiments as the percent of litter mass remaining (LMR %) calculated as the percentage of the dry
mass of each type of litter (decomposing alone or in the mixture) at the end of the experiment concerning its respective initial dry
mass accordingly to the following Eq. (1):

$$LMRi\ (\%) = \left(\frac{Fdwi}{Idwi}\right) \times 100, \tag{1}$$

where Fdw$i$ and Idw$i$ are the final and initial dry weights of litter $i$ (flowers or leaves), respectively.

228       To estimate the total percentage of litter mass remaining for the two litter types combined in the observed mixture, we used

Eq. (2):

$$LMR_{obs}(\%) = \left(\frac{Fdwf+Fdwl}{Idwf+Idwl}\right) \times 100, \tag{2}$$

where Fdw$f$ and Fdw$l$ are the final dry weights of flower and leaf litter, respectively, and Idw$f$ and Idw$l$ are the initial dry weights
of flower and leaf litter at the beginning of the experiment, respectively.

233       To quantify the LMEs for the whole mixture, we compared the observed ($LMR_{obs}$) and the expected ($LMR_{exp}$) LMR (Loreau,

1998). The expected LMR for the combined responses of both litter types to litter mixing was calculated using Eq. (3) assuming
no interaction between both litter types:

$$LMR_{exp}(\%) = \left(\left(LMR_f\right) \times (pi)\right) + \left(\left(LMR_l\right) \times (pi)\right), \tag{3}$$

where LMR$_f$ is the percentage of flower litter mass remaining in the monoculture, LMR$_l$ is the percentage of leaf litter mass
remaining in the monoculture, and *pi* is the proportion of the biomass of litter *i* in the mixture.
Then, we calculated the relative mixture effect (RME) in each litter type for each ecosystem (Barantal et al., 2011) using
Eq. (4):
$$RME\ (\%) = \left(\frac{LMR_{obs} - LMR_{exp}}{LMR_{exp}}\right) \times 100, \qquad (4)$$
where RME is the relative mixture effect (%) for the whole combined litter or each litter type in the mixture; for the whole mixture,
LMR$_{exp}$ is the expected litter mass remaining calculated by averaging the LMR values of both litter types in monoculture, and
LMR$_{obs}$ is the observed litter mass remaining of the whole mixture calculated by averaging the observed LMR values of both litter
types in the mixture. For each litter type, LMR$_{exp}$ is the flower or leaf litter mass remaining in the monoculture and LMR$_{obs}$ is the
flower or leaf litter mass remaining in litter *i* in the mixture. For RMEs, positive and negative values indicate that litter decomposes
faster and slower in mixtures than in its respective monoculture, respectively.

**2.5 Data analysis**
To test the functional differences between the litter types of *T. aurea*, we compared the average concentrations of functional
traits (C, Cel, Lig, Phe, S-Carb, N, P, K Na, Mg, Ca, WHC, leaching, and some stoichiometric ratios (Lig:S-Carb, Lig:N, Lig:P)
between both litter types using an unpaired t-test.

253          We conducted a set of statistical analyses to test our hypotheses. Initially, we conducted unpaired t-tests between the litter

types to evaluate if the decomposition rate of flower and leaf litter differed. Specifically, we compared the LMR of each litter type
in monocultures in both terrestrial and aquatic experiments. Then, we conducted regression analyses to assess the effect of flower
litter biomass on the decomposition of each litter type, alone and in mixtures. Specifically, we assessed the LMR of each litter type
in monocultures and mixtures as a function of the variation in flower litter biomass proportion. Next, we employed the test of
heterogeneity of slopes to determine if litter mixture affected the biomass-decomposition relationship of each litter type separately
as well as to the mixture. For this analysis we considered the proportion of flower litter biomass in the litter mixture as the predictor
and the LMR of each litter type alone or combined as the response variable. Statistically significant effects (i.e. when the slopes of
the regressions are different from each other) would indicate that the biomass-decomposition relationship differed between litter
decomposing alone and in mixture in response to flower litter biomass proportion. This method is equivalent to an Analysis of
Covariance (ANCOVA) (Zar, 1984). In cases where the slopes of the regressions did not differ significantly from each other, we
used unpaired t-tests to compare the grand mean decomposition (i.e. irrespective of litter biomass) of each litter type alone as well
as for the whole mixture. The aforementioned analytical procedures were performed separately for terrestrial and aquatic
experiments because of experimental design differences between them (see details in *Litter sampling and experimental design and*
*setup*). To assess whether the RME for each type of litter and the mixture is a function of their respective biomass in the mixture,
we utilized linear regressions for both aquatic and terrestrial experiments.
Before linear regression analysis, the data were tested for assumptions of linearity with Run's Test. The homogeneity of the
residuals as assumptions for linear regressions and unpaired t-tests were tested using the Bartlett Test. All statistical analyses were
performed using the GraphPad Prism software (version 6.0). A level of significance of $\alpha = 0.05$ was considered for all analyses.

**3 Results**
**3.1 Flower and leaf litter chemical and structural composition**
Overall, flower litter had a more labile chemical composition and physical traits than leaf litter. Except for the C content,
all analyzed chemical constituents displayed significantly different concentrations between the litter types (Fig. 1a). Specifically,

flower litter had significantly higher concentrations of S-carb, N, P, and K than leaf litter, whereas leaf litter had significantly higher concentrations of Cel, Lig, Phe, Ca, Mn, and Na than did flower litter (Fig. 1a). Additionally, the Lig:S-Carb ratio, which indicates the relative proportion of recalcitrant and labile C, was significantly lower in the flower litter than in the leaf litter. The same pattern was observed for Lig:N and Lig:P ratios (Fig. 1b). Finally, the physical traits followed the same pattern, with flower litter exhibiting higher values of WHC and leaching than leaf litter (Fig. 1c).

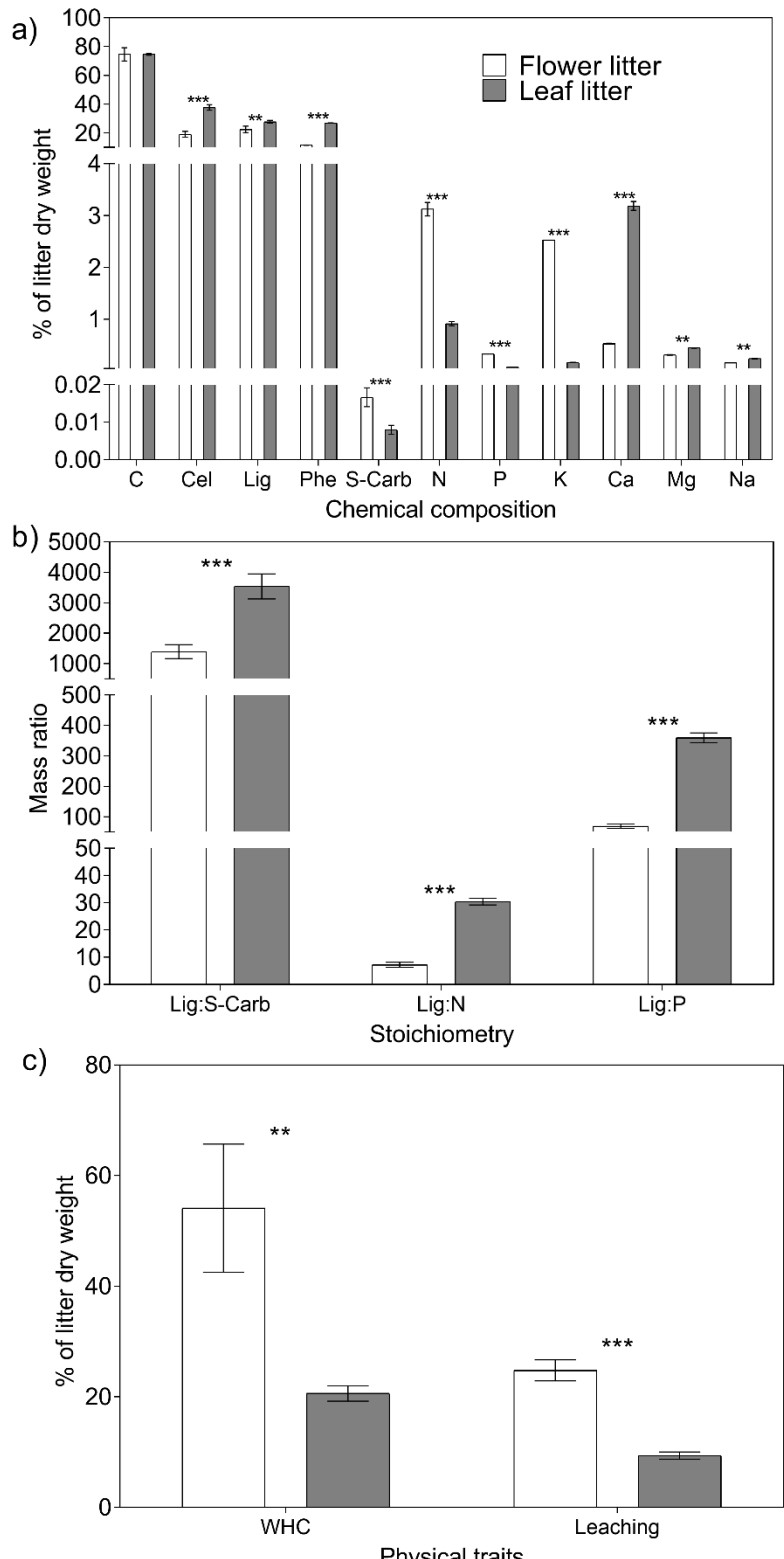

**Figure 1:** Average (n=3 ± 1SD) functional trait values for a) concentration of chemical constituents, b) stoichiometry, and c) physical traits for flower and leaf litter of *Tabebuia aurea*. Abbreviations are as follow: Cel – cellulose, Lig – lignin, Phe – phenolic compounds, S-carb – soluble carbohydrates, WHC – water holding capacity. Asterisks above the bars indicate significant statistical differences between the flower and leaf litter for the respective functional traits (unpaired t-test; **p<0.001; ***p<0.0001).

### 3.2 Differences in the decomposition rate between the litter types in monoculture

Considering only the values of leaf and flower litter decomposing in monocultures, the leaf litter decomposed significantly slower than flower litter in the terrestrial experiment, with the average leaf LMR (84.4%) being significantly higher compared to the average flower LMR (41.8%) (Fig. 2ab; t=72.4; p<0.0001; unpaired t-test). This pattern was consistent with that observed in the aquatic experiment, and the values of leaf and flower litter decomposing in monoculture were significantly different from each other. The leaf litter decomposed significantly slower than flower litter, with average leaf LMR (72.7%) significantly higher than the average flower LMR (50.3%) (Fig. 3ab; t=13.3; p<0.0001; unpaired t-test).

### 3.3 Litter-mixing effects of flower and leaf litter on decomposition in the terrestrial experiment

Leaf litter decomposition rates, expressed as LMR, did not significantly vary across the gradient of leaf litter biomass in the monoculture (Fig. 2a). However, leaf litter decomposition was significantly altered when mixed with flower litter. In general, an increase in the amount of flower litter had significant, positive (i.e. lower LMR values) and linear effects on leaf litter decomposition rates (Fig. 2a; $F_{1, 14}$ = 215.9; p<0.0001). However, interestingly, in the two mixtures with the lowest flower:leaf litter biomass proportion, the decomposition of leaf litter were lower than those observed in their respective monocultures (i.e. higher LMR values; Fig. 2a).

Similar to leaf litter, flower litter decomposition rates did not vary significantly in response to its biomass variation in the monoculture (Fig. 2b). However, as observed for leaf litter decomposition, flower litter decomposition was significantly altered when mixed with leaf litter. In this scenario, the increasing amounts of leaf litter in mixtures had significant but negative (i.e., higher LMR values) and linear on flower litter decomposition rates (Fig. 2b; $F_{1, 14}$ = 65.9; p<0.0001).

Finally, we observed similar significant effects on the variation in LMR for both litter types combined in response to the proportion of flower litter mass in the mixture (Fig. 2c). Decreasing the amount of flower litter in the mixture significantly increased both the expected and observed values of LMR for the litter mixture as a whole, although the slopes of both relationships did not differ significantly (Fig. 2c; $F_{1, 14}$ = 3.5; p=0.08). Litter-mixing effects on the whole mixture was significant (Fig. 2c – right panel; t=16.5; p<0.0001 unpaired t-test), as the average observed LMR for the whole mixture (37%) was significantly lower than its average expected value (63%) calculated from the decomposition of both litter types alone. These results indicated that, on average, litter mixing had a stimulating effect of 26% on the decomposition of the whole mixture treatment.

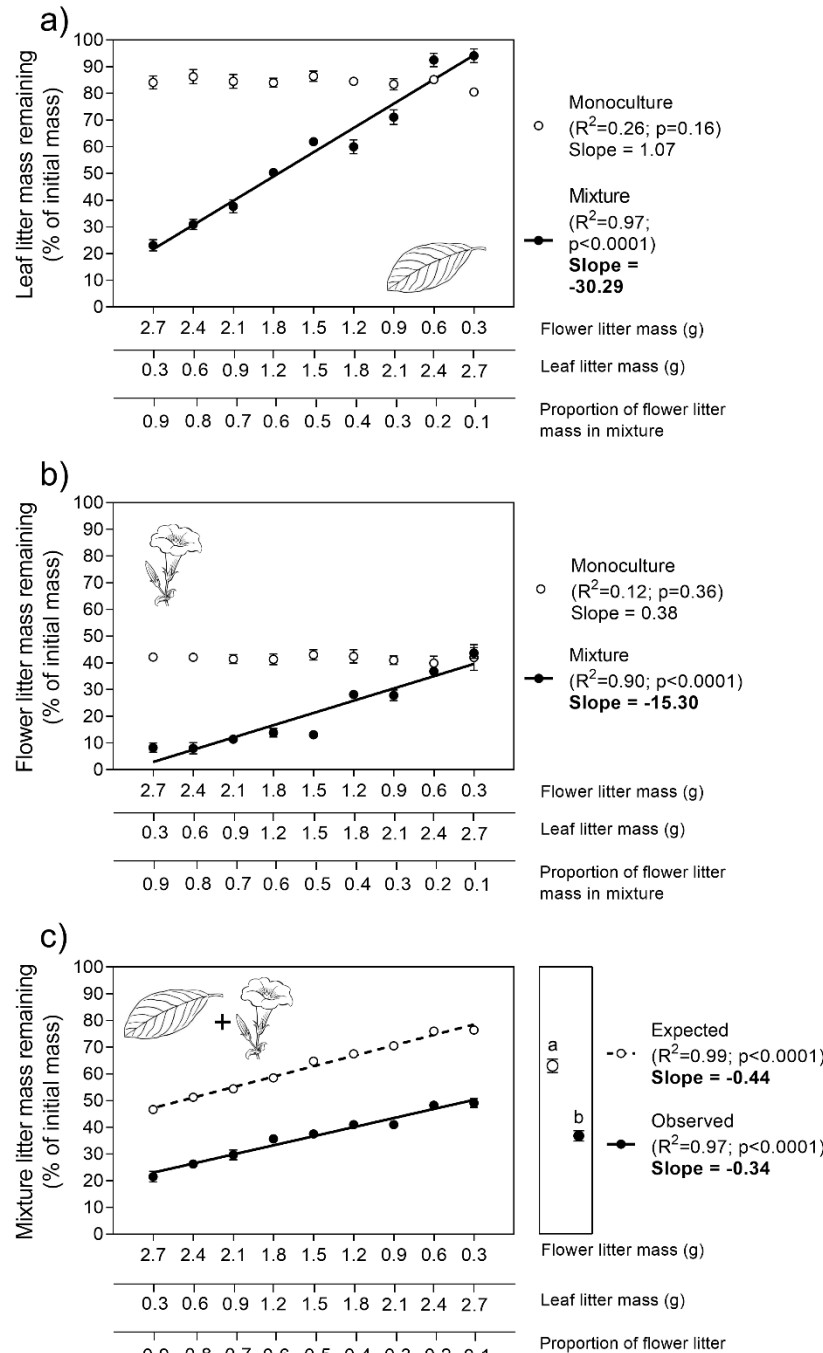

**Figure 2:** Patterns of flower and leaf litter mass remaining in the terrestrial experiment for single a) leaf, b) flower, and c) whole mixture (leaf +flower) decomposing alone (i.e. monocultures) or mixed (i.e. mixtures). Litter mass remaining for each litter type alone and in combination were fitted as linear functions of flower:leaf litter proportion. Values in left panels are mean (n=10 ± 95% CI). Slope values in bold indicate significant statistical differences regarding the interactive effects between the explanatory variable and litter mass remaining (p<0.05; F-test for Homogeneity of slopes analysis). Values in the right panel of Fig. 2c depict the grand mean (n=90 ± 95% CI). Different letters above the grand mean values indicate significant statistical difference (p<0.05; unpaired t-test).

**3.4 Litter-mixing effects of flower and leaf litter on decomposition in the aquatic experiment**

Leaf litter biomass did not significantly affect the variation in leaf LMR, either alone or in combination with flower litter (Fig. 3a; $F_{1, 14} = 0.1$; p=0.76). However, when mixed with flower litter, the average leaf LMR was significantly lower (69.5%) than that of its monoculture (72.7%), indicating that leaf litter decomposed 3.2% faster on average in the presence of flower litter (Fig. 3a, right panel; t=2.1; p=0.04, unpaired t-test).

Similar patterns were observed for flower litter decomposition. Variations in flower litter biomass had no effect on flower LMR, either alone or in combination with leaf litter (Fig. 3b; $F_{1, 14} = 0.3$; p=0.62). However, when mixed with leaf litter, the average flower LMR was significantly lower (47.1%) than that of its monoculture (50.3%), indicating that flower litter decomposed on average 3.2% faster in the presence of leaf litter (Fig. 3b, right panel; t=2.2; p=0.04, unpaired t-test).

Finally, the expected and observed values for the LMR of the whole mixture increased significantly as a function of the decreasing proportion of flower litter mass in the mixture (Fig. 3c). However, similar to the observations in the terrestrial experiment, the slopes of both relationships were not significantly different from each other (Fig. 3c; $F_{1, 14} = 1.3$; p=0.27). Nevertheless, the LMEs on the decomposition of the whole mixture were significant (Fig. 3c, right panel; t=2.3; p=0.03, unpaired t-test), as the average observed LMR for the litter mixture (57.5%) was significantly lower than its expected value (61.5%), calculated from the decomposition of both litter types alone, indicating that litter mixing had, on average, a stimulating effect of 4% on the decomposition of the whole litter mixture.

Litter-mixing effects on the whole mixture was significant (Fig. 2c – right panel; t=16.5; p<0.0001 unpaired t-test), as the average observed LMR for the whole mixture (37%) was significantly lower than its average expected value (63%) calculated from the decomposition of both litter types alone.

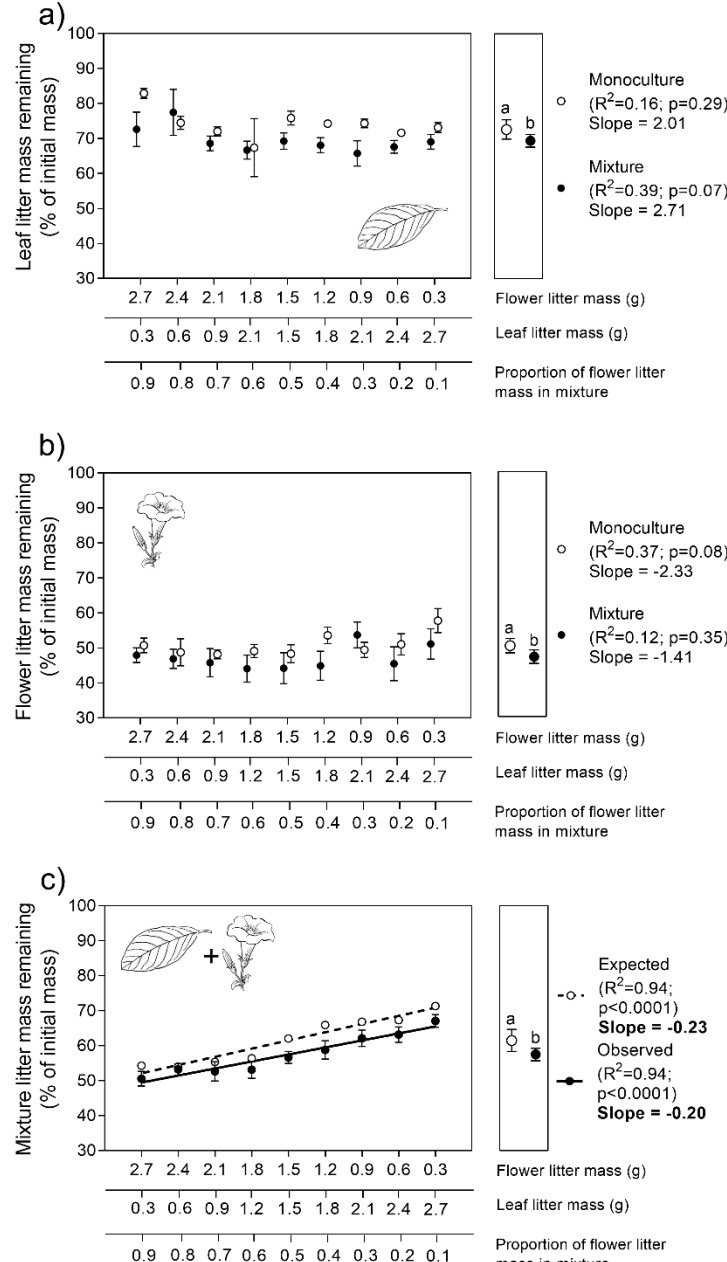

Figure 3: Patterns of flower and leaf litter mass remaining in the aquatic experiment for single a) leaf, b) flower, and c) whole mixture (leaf +flower) decomposing alone (i.e. monocultures) or mixed (i.e. mixtures). Litter mass remaining for each litter type alone and in combination were fitted as linear functions of flower:leaf litter proportion. Values in left panels are mean (n=3 ± 95% CI) for monocultures and (n=6 ± 95% CI) for mixtures. Slope values in bold indicate significant statistical differences regarding the interactive effects between the explanatory variable and litter mass remaining (p<0.05; F-test for Homogeneity of slopes analysis). Values in right panels depict the grand mean (n=27 ± 95% CI) for monocultures and (n=54 ± 95% CI) for mixtures. Different letters above the grand mean values indicate significant statistical difference (p<0.05; unpaired t-test).

**3.5 The magnitude of RME for terrestrial and aquatic experiments**

Variations of RME values for the two litter types and for the whole litter mixture in response to flower:leaf litter biomass
proportion showed distinct patterns for terrestrial and aquatic experiments (Fig. 4). In general, RME values were higher for the
terrestrial experiment compared to the aquatic experiment considering each litter type and the whole mixture. In the terrestrial
experiment, RME values for flower, leaf, and both litter types combined, varied significantly and positively as a function of
flower:leaf litter proportion (Fig 4a). However, the variation of RME values for flower and leaf litter as a function of flower:leaf
biomass proportion were parallel and not statistically different from each other ($F_{1, 14} = 0.04$; p=0.85) but they were both statistically
different (steeper) from the variation of RME values for the whole mixture ($F_{2, 21} = 21.2$; p<0.0001). On average, the decomposition
of flower litter increased 49% in the presence of leaf litter and the decomposition of leaf litter increased, on average, 31.2% in the
presence of flower litter, but such a difference was not statistically significant (Fig. 4a right panel; t=1.2; p=0.25 unpaired t-test).
Contrary to what was observed in the terrestrial experiment, we did not observe significant effects of flower:leaf litter
biomass proportion on the variation of RME values for leaf and flower litter as well as for whole mixture in the aquatic experiment
(Fig. 4b). Furthermore, average values of RMEs for flower (7%) and leaf litter (6%) were not significantly different from each
other (Fig. 4b right panel; t=0.34; p=0.74 unpaired t-test).

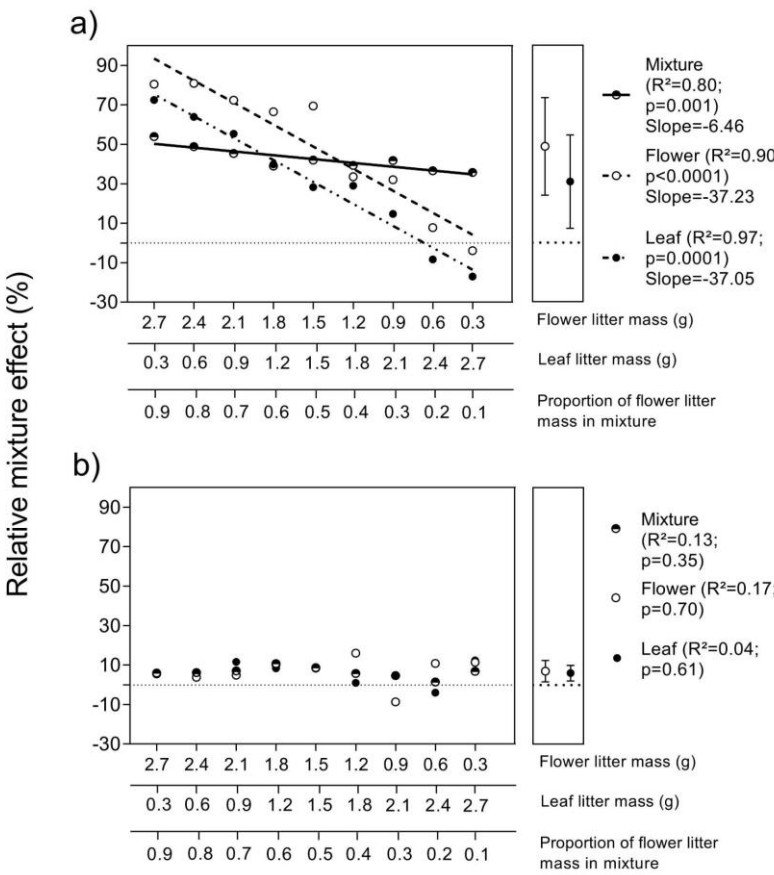


**Figure 4:** Response of the relative mixture effect (RME) of flower, leaf and whole mixture to the gradient of flower:leaf litter
proportion in a) terrestrial and b) aquatic ecosystems. Regression lines sided by different letters indicate their slopes are statistically
different from each other (p<0.05; F-test for Homogeneity of slopes analysis). Values in the right panel of Fig. 4a depict the grand
mean. Symbols represent mean (n=9 ± 95% CI). Error bars were shorter than the size of the symbols. The ± 95% CIs values were
calculated via bootstrap.

**4 Discussion**

Our study is the first to assess the LMEs of flower and leaf litter mixture on decomposition across terrestrial and aquatic ecosystems. Our findings suggest that flowers have a lasting interactive effect on litter decomposition beyond their role in reproduction, providing evidence of LMEs resulting from the mixture of litter from different plant organs, even at the intra-specific level. First, our findings reveal that *T. aurea*'s flower and leaf litter have distinct functional traits and decomposition rates. Consistent with our first hypothesis, *T. aurea*'s flower litter exhibits chemical and physical functional trait values indicative of a more labile detritus compared to *T. aurea*'s leaf litter. Such functional divergences between flower and leaf litter quality were confirmed by the higher decomposition rates of flower litter compared to leaf litter in both terrestrial and aquatic environments. Secondly, our results strongly supported that the interaction between flower and leaf would result in LMEs since the occurrence of LMEs of the flower and leaf litter mixture were consistent in terrestrial and aquatic environments. Third, we hypothesized that the LMEs of flower litter on leaf decomposition would be stronger and more positive than those from leaf litter on flower litter. However, flower and leaf litter mixing had reciprocal effects on the decomposition of each other, with symmetric LMEs (both in magnitude and direction) on the decomposition of both litter types and in both terrestrial and aquatic ecosystems. Finally, our fourth prediction that the LMEs resulting from the mixture of flower and leaf litter would vary in magnitude and direction depending on the proportion of flower and leaf biomass in the litter mixture, was supported only in the terrestrial environment, where the decomposition rate of leaves, flowers, and the whole mixture, was faster with increasing proportion of flower litter in the litter mixture. This result has two important ramifications. A more specific one suggests that the unbalance distribution of nutrients in flower and leaf tissues has consequences on litter decomposability after the senescence process, with further effects to the interaction between flower and leaf litter decomposition (Schmitt and Perfecto, 2020). This result expands what have been reported by recent studies showing the effects of litter mixing from different plant organs for the occurrence of LMEs on decomposition (de Paz et al., 2018; Hou and Lü, 2021). Secondly, a more general implication of our results indicates that since the increasing proportion of flower litter in the mixture increased the decomposition of litter mixture as a whole, the LOM:ROM biomass proportion in the detritus pool can be a crucial factor mediating the mechanisms controlling the decomposition process in ecosystems (Guenet et al., 2010). Finally, our results also reinforce the notion of what has been observed in studies that seek to synthesize the effects of litter functional diversity on decomposition, which advocate that LMEs are more consistent in terrestrial than in aquatic ecosystems (Gessner et al., 2010) .

Leaves and flowers are plant organs with distinct functions and forms, resulting in differences in their chemical and physical characteristics. Recent research has emphasized the significance of indirect effects of ecological and evolutionary mechanisms in shaping litter decomposition through legacy effects from functional traits of living plant tissues that persist after tissue death and impact litter decomposition through after-life effects (Freschet et al., 2012; Cornelissen et al., 2023). The longer lifespan and persistence of leaves require the plant to invest more in structural tissues to provide greater physical resistance for these organs, as well as in secondary compounds that act as constitutive defenses against herbivory and photo-damage by UV radiation (Stamp, 2003). Additionally, leaf senescence is generally slow, and many species have efficient nutrient resorption prior to leaf abscission, especially deciduous species living in infertile soils (Brant and Chen, 2015) like the one used in our study. On the other hand, flowers are ephemeral reproductive organs with high concentrations of nutrients and soluble compounds, as well as labile C compounds that make up nectar to attract pollinators (Freeman et al., 1991; Galetto and Bernardello, 2004). Compared to leaves, flowers are generally fast-growing and short-lived organs (Ashman and Schoen, 1994), which results in higher nutrient investment, lower nutrient resorption and lesser herbivore attack than leaves (Mccall and Irwin, 2007). As a result, flower litter may contain higher concentrations of nutrients and labile C compounds, while having lower amounts of structural tissues and

deterrent secondary compounds (Stamp, 2003). These conjectures were most supported by our data and confirmed our hypothesis that flower litter is a more LOM than leaf litter and decomposes faster than leaf litter in both terrestrial and aquatic ecosystems. Flower litter has higher concentrations of N, P, K, and labile C, as well as higher leaching capacity and WHC, while leaf litter were richer in micronutrients such as Mg, Ca, and Na. These results also supported the assumptions we have used to rationalize our second hypothesis, which predicted that due to functional differences in litter quality, the mixture of flower and leaf litter would cause LMEs on the decomposition of both litter types. Studies in both terrestrial and aquatic ecosystems have shown that the litter functional dissimilarity rather than litter species number is the most important factor causing LMEs on decomposition (Epps et al., 2007; Lecerf et al., 2011; Violle et al., 2017). Our study supports this paradigm in demonstrating that LMEs on decomposition can also occur intraspecifically via the interaction of flower and leaf litter, and call attention to the importance of LMEs on decomposition even in low-diverse systems through the interactions of litter from different plant organs.

Meanwhile, according to our third hypothesis, we expected the LMEs between flower and leaf litter to be asymmetrical, with the leaf litter decomposing more quickly in the presence of flower litter than vice versa. Our conjecture relied on the results of past studies that show that litter with contrasting qualities affects and/or responds to litter mixing in distinct ways. In general, studies have shown that LOM stimulates the decomposition of ROM mainly due to nutrient transfer and/or priming effects (Guenet et al., 2010; Liu et al., 2020), while ROM may inhibit the decomposition of LOM due to the presence of deterrent secondary metabolites (Hättenschwiler et al., 2005). However, our findings rejected our third prediction. While we observed that, on average, flower litter (LOM) accelerated the decomposition of leaf litter (ROM), we also found that leaf litter reciprocally accelerated the decomposition of flower litter, and the magnitudes of these effects were statistically indistinguishable from each other in both terrestrial and aquatic experiments. These results are due to complementary effects. The two litter types have highly contrasting chemical and physical characteristics. Flower litter has a higher concentration of labile C and nutrients (N, P, K), and WHC and leaching potential than leaf litter. This pattern points to the possibility of flower litter to accelerate the decomposition of leaf litter through mechanisms such as nutrient transfer and/or improved microenvironment conditions (Hättenschwiler and Jørgensen, 2010; Makkonen et al., 2013). On the other hand, leaf litter has higher concentrations of micronutrients such as Mg, Ca, Na, which could, via micronutrient transfer, compensate for possible limitations of flower litter decomposition by these elements. In fact, studies have already shown that litter decomposition is co-limited by macro and micronutrients in tropical forests (Kaspari et al., 2008; García-Palacios et al., 2016). Additionally, specifically for the terrestrial experiment, the higher toughness of leaf litter may have influenced microenvironmental conditions of the litter layer inside microcosms (Makkonen et al., 2013), preventing the compaction of the flower litter layer and avoiding anaerobic conditions, which could negatively affect the decomposition of flower litter in monocultures.

However, for a better understanding of the aforementioned mechanisms underlying the LMEs of flower and leaf litter mixing on decomposition, it is essential to consider how these effects, respective to each litter type, varied in response to the flower-to-leaf litter proportion. In the terrestrial experiment, we observed that RME of leaf litter increased in response to an increase in flower litter biomass in the litter mixtures, and negative RME values for leaf litter decomposition were observed only in the two mixtures with the lowest flower litter biomass. As discussed above, the variation in positive values of RME in response to the increasing biomass of flower litter in mixtures points out to mechanisms that are generally attributed to the enhancing effects of LOM on the decomposition of ROM in litter mixtures, such as nutrient transfer or mining and priming effects (Guenet et al., 2010; Chen et al., 2014). The resource concentration hypothesis posits that resource quantity drives resource use efficiency (Charnov, 1976; Hambäck and Englund, 2005). The optimal foraging efficiency of microbial decomposers depends on mechanisms that maximize the balance between enzyme production and energy gain. We conjectured that the increase in flower litter biomass could

have optimized the enzyme production and energy for the maintenance of metabolic processes, known as the substrate induction hypothesis (Allison et al., 2014; Schimel and Weintraub, 2003), which might have enhanced the leaf litter decomposition. Another possibility is the occurrence of priming effects mechanisms, such as co-metabolism, which posits that the decomposition of ROM may be enhanced by LOM targeting enzymes capable of degrading the ROM, and/or that LOM decomposition may supply energy for microorganisms to produce extracellular enzymes capable of degrading ROM (Guenet et al., 2010). Such LMEs, resource-mediated mechanisms of flower litter on leaf litter decomposition in the terrestrial experiment may have acted in combination with the improvement of microenvironmental conditions promoted by the higher WHC of flower litter. Otherwise, the antagonistic effects observed in treatments with lower biomass of flower litter may be associated with the preferential feeding of decomposers on flower litter, and the low energy provide by the LOM was not enough to induce the degradation of the ROM (Cheng, 2009; Wang et al., 2015). It is important to note that we did not use labeled material to clearly distinguish the ROM and the LOM dynamic as classically done in priming experiments. Therefore, our priming related interpretation must be taken with due care.

However, what could explain the unexpected variation of flower litter decomposition in the terrestrial experiment as the proportion of flower-to-leaf litter varied in litter mixtures? Although the decomposition of flower litter was enhanced in the presence of leaf litter irrespective of the flower-to-leaf litter proportion, these effects consistently weakened as the amount of leaf litter increased and the amount of flower litter decreased in the litter mixture. We conjectured that such results might have occurred due to the combination of two potential mechanisms. First, the presence of even a small amount of leaf litter could have an enhancing effect on the decomposition of flower litter if it is enough to meet the microbial decomposer community's demand for a specific limiting nutrient in the mixture. This, for example, might have occurred in litter mixtures due to the higher concentrations of micronutrients such as Ca, Mg, and Na, in leaf litter compared to flower litter. These micronutrients are considered important for litter decomposition in tropical forests (Kaspari et al., 2009), and their demand for decomposers is comparatively lower than macronutrients, such as N and P (Tyler, 2005). Therefore, even the low proportion of leaf litter might have been sufficient to meet the micronutrient demand of decomposers for decomposing flower litter. Secondly, the interaction between different types of litters can affect their decomposition through two non-mutually exclusive ways: through the effect a given litter can have on the other and/or through the response a given litter can exhibit to the interaction with another litter in the mixture. For example, labile litter can both expedite the breakdown of another litter (as discussed earlier), but also it may be more reactive to stimulation from another litter. This is because labile litter typically offers fewer resources that limit decomposers. Consequently, when a stimulus results from interactions with another litter, it's more likely to boost the decomposition of labile litter compared to refractory litter. We believe that this mechanism may have been relevant in determining the observed results in the terrestrial experiment, as the synergistic effects of the mixture of litter on flower litter decomposition increased with the rise in flower litter biomass in the mixture, while decreased with the increasing in the amount of leaf litter in the mixture.

The patterns resulting from the mixture of flower and leaf litter and the variation in the relative biomass of these litter types in the mixture were much less pronounced in the aquatic experiment. Although litter mixing also resulted in synergistic effects in the aquatic experiment, such effects were weaker compared to those observed in the terrestrial experiment and did not consistently vary with the variation in the flower-to-leaf litter biomass proportion. Overall, these results support commonly reported observations in the literature regarding the effects of detritus mixture on decomposition, showing that LMEs are generally weaker or absent in aquatic ecosystems compared to terrestrial ones (Gessner et al., 2010). These results may be, in their entirety or in part, attributed to the fact that mechanisms potentially relevant for triggering LMEs in terrestrial ecosystem, such as moisture exchange between different types of litters and LMEs on physical properties of litter layer, lose relevance in the aquatic environment (Schmidt et al., 2011; Bengtsson et al., 2018). However, we may also have underestimated the LMEs in the aquatic experiment since, in this environment, a considerable portion of organic matter is leached from the litter and degraded in the water

column in its dissolved form, which was not quantified in our experiment. In fact, our results showed that leaching is responsible for causing 25% and 9% of mass loss on flower and leaf litter, respectively (Fig 1c). Several studies have demonstrated that litter-mixing interactions between dissolved organic matter from litter of different qualities also occur and accelerate decomposition in the water column (Farjalla et al., 2009; Fonte et al., 2013). Hence, forthcoming studies should explore the impact of flower and leaf litter mixture on both particulate and dissolved organic matter decomposition to achieve a more comprehensive understanding of the LMEs of flower and leaf litter on decomposition.

Although the results of our experiment have demonstrated consistent patterns of flower and leaf litter mixture in the occurrence, magnitude, and direction of LMEs in decomposition, it is important to consider some caveats of our experiment. Although we made an effort to maintain environmental conditions similar to those observed in nature, laboratory conditions will always suppress features of the environment that may be relevant to the study at hand. Firstly, we were unable to measure how the effect of the mixture and the variation in the proportion of flower and leaf litter affected the microbial community, which was the primary group of decomposers mediating our results. Secondly, the absence of macrofauna in our experiment could limit an accurate estimation of LMEs through flower and leaf litter interaction, since the presence of macro-fauna has been repeatedly shown to be an important factor in determining the occurrence and magnitude of synergistic LMEs on decomposition through litter fragmentation and decomposers complementary use of litter resources (Hättenschwiler and Gasser, 2005; Njoroge et al., 2022, 2023). Therefore, in future studies the inclusion of macrofauna could be important to quantify the real impact of flower and leaf litter interaction in nutrient dynamics in ecosystems. Thirdly, in the aquatic experiment, we simulated stillwater environmental conditions typically observed in lentic systems, such as temporary pools along the channel of intermittent rivers, small ponds, phytotelmata, and so forth. These environments are generally nutrient-poor and result in the prolonged confinement of water and litter (Migliorini et al., 2018; Bonada et al, 2020), potentially affecting the generalizability of our results to other aquatic systems. However, our incubation method may not fully replicate real-world conditions, especially within lotic ecosystems. Therefore, future studies should assess the occurrence, magnitude, and direction of LMEs resulting from the interaction between flower and leaf litter in lotic systems. In these systems, there is a long tradition of studies evaluating the decomposition of detritus from riparian vegetation, yet the importance of the interaction between leaf and floral litter in decomposition is often overlooked (but see Rezende et al., 2017).

Massive flowering is a phenology pattern found in a range of species in different ecosystems, beyond the Bignoniaceae family (Conceição et al., 2013; Whigham, 2013; Zheng et al., 2020). The litter-mixing interactions between flower and leaf litter could be widespread in natural ecosystems, caused by differences in quality between flowers and leaf litter. These differences may be primarily attributed to variations in the form and function of these organs, creating a legacy effect for decomposition (Freschet et al., 2013; Cornelissen et al., 2023). In particular, our results indicate that species that present massive flowering phenology may represent key roles mediating the occurrence of temporal and/or spatial biogeochemical hotspots (Kuzyakov, 2010), both through direct effects, where the flowers themselves generally represent a more labile and rapidly decomposing litter, thus being recycled more quickly and efficiently, and through indirect effects, where flower litter can interact with leaf litter complementarily stimulating the decomposition of both litters at the litter layer around the flowering tree. However, the results observed in the terrestrial experiment, which highlight that the magnitude of LMEs depends on the flower-to-leaf litter biomass proportion in the litter mixture, may represent the occurrence of a phenomenon analogous to the Janzen-Connell Hypothesis (Janzen, 1970; Connell, 1971). This hypothesis predicts that patterns of seed predation (Janzen, 1970) and herbivory on seedlings (Connell, 1971) are more intense near the parent tree because of resource concentration effects. In forests, the interaction between flower and leaf litter could occur beyond the taxonomic level, if differences of litter quality between flower and leaf litter were widespread. For example, in

dense forests the presence of a few scattered trees presenting massive flowering can generate LMEs on leaf litter at either intra- or
inter-specific levels (Fig. 5a). Analogously to what is expected for seed and seedlings distribution according to the Janzen-Connell
Hypothesis, flower litter biomass should be more concentrated below the flowering tree and gradually decrease farther from it. On
the other hand, leaf litter biomass would be more homogeneously distributed in the litter layer due to the contribution from the
neighboring trees (Fig. 5b). This differential input of flower and leaf litterfall to the litter layer could generate a pattern of variation
both in the concentration of flower litter and in the proportion of flower-to-leaf litter in the litter layer in relation to the blooming
tree. Therefore, the rate of nutrient recycling due to decomposition should be higher near the flowering tree due to the
decomposition of the high flower litter biomass itself and because, according to our results, synergistic effects of flower and leaf
litter on the decomposition of both litter types are stronger in high flower-to-leaf litter biomass proportions (Fig 5b). These potential
effects would be more important in terrestrial ecosystems, both because the LMEs of litter mixing were responsive to variation in
the flower-to-leaf litter biomass proportion only in the terrestrial experiment and because the arrangement of higher flower litter
concentration around the flowering tree is more likely to be found in terrestrial ecosystems. However, the conjecture presented in
this conceptual model must be properly tested to validate its expected results since our experiment, although allowing us to
speculate on potential hypotheses and patterns, does not enable us to test or confirm them. Due to the importance of flower:leaf
biomass to the occurrence of LMEs, future studies should quantify the flower:leaf biomass proportion in natural conditions to
accurately understand the effects of flower on litter decomposition and which flower:leaf litter biomass proportions often generate
LMEs.

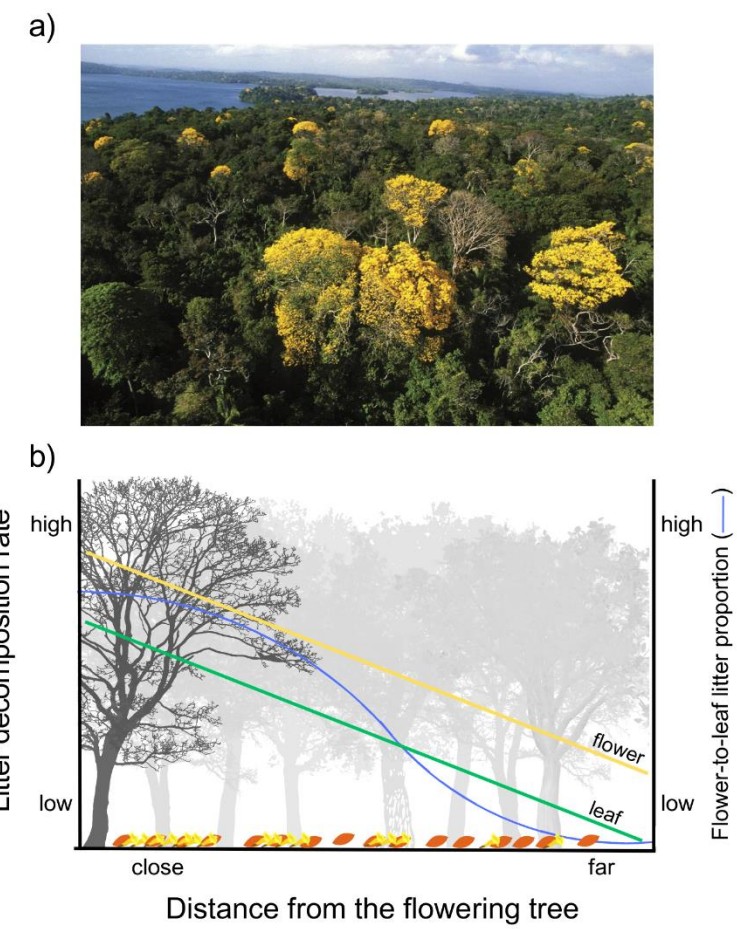


**Figure 5:** Conceptual model expanding the importance of flower and leaf litter spatial distribution and potential interactions in
relation to the distance of the flowering tree in natural forests. a) Scattered distribution of massive flowering trees in a natural
forest; Photo by S. Joseph Wright, Smithsonian Tropical Research Institute. b) We adapted a theoretical scheme based on the

Janzen-Connell Hypothesis, which assumes that predation on seed and herbivory on seedlings decreases along the distance from the parent tree as seeds and seedlings become rarer on the forest floor. In our case, we assume that nutrient recycling, measured as litter decomposition rate, is a function of the absolute and relative biomass of flower and leaf litter in the litter layer. Absolute and relative biomass of flower litter decreases along the distance from the flowering tree. This is because the dispersion of flower litter to the litter layer is stronger near the flowering tree, becoming increasingly weaker with distance from the flowering tree, and the quantity of leaf litter from all neighboring tree species in the litter layer is independent of the distance from the flowering tree. Near the flowering tree, recycling through decomposition is expected to be higher due to the large amount of flower litter, which decomposes quickly because of its high quality, but also because the litter-mixing effects of the interaction between flower and leaf litter are stronger in the litter mixtures with a high proportion of flower-to-leaf litter. The results used to conjecture the predictions of this conceptual model are presented in Fig.4a.

**5 Conclusions**

Our findings highlight the importance of litter from plant reproductive organs for LMEs in ecosystems, which could substantially contribute to changes in nutrient and carbon dynamics. Our results highlight the importance of intra-specific variability among organs indicating the occurrence of LMEs could be more dependent on litter dissimilarity than taxonomic richness, suggesting the potential relevance of LMEs at intra-specific levels in low-diversity communities. Although recent studies have evidenced the role of reproductive organs in increasing the decomposition of organic matter in the natural environment in both terrestrial (de Paz et al., 2018; Schmitt and Perfecto, 2020) and aquatic (Rezende et al., 2017) ecosystems, it is necessary to evaluate the general patterns and the possible mechanisms driving the effects of litter from reproductive organs on LMEs in each environment. We found evidence of the central role of complementarity mechanisms in the occurrence, magnitude, and direction of LMEs. In the terrestrial environment, the LMEs varied as a function of flower-to-leaf litter biomass proportion in the litter mixture, indicating the potential importance of interplay between resource quality and quantity in determining niche partitioning among microbial decomposers. To understand the generalities of the second role of flowers on litter decomposition, it is important to verify the generality of our results found for *T. aurea*. Future studies should investigate the generalities of flower and leaf litter on LMEs at intra- and inter-specific levels, as well as the potential role of flower litter in affecting direct and indirect mechanisms of LMEs on litter decomposition across a large variety of plant species, an aspect that has been completely neglected in the literature about the effects of litter mixing on decomposition.

**Code availability**

Code is available upon request from the corresponding author.

**Data availability**

Data is available upon request from the corresponding author.

**Authors contribution**

MIGA was responsible for running the experiments and writing the original draft. RDG, LSC, and AC were responsible for conceptualizing the experiments. BG and AC were responsible for the funding acquisition. MIGA and AC were responsible for the formal analyses. BG, MIGA, and AC were responsible for data interpretation. ELV was responsible for providing material and reagents for the chemical analyses. All co-authors contributed by reviewing and editing the submitted version.

**Competing interests**

At least one of the (co-)authors is a member of the editorial board of Biogeosciences.

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
