# Peer review of ""Blooming" of litter-mixing effects: The role of flower and leaf litter"

_EGUsphere, 2024_

## Referee Comment (RC2)

This manuscript presents the effects of varied proportions of leaf and flower litter on litter-mixing effects in terrestrial and aquatic ecosystems. The study design and findings are interesting, particularly the investigation on intra-specific interactions on decomposition, which I found to be a novel contribution to the field. I believe the study presents valuable insights and will inspire further research in this area. Given the important role of litter-mixing effects in nutrient and carbon dynamics, the manuscript fits well within the scope of BG.

However, I have a concern regarding the limitations of the study. While the authors acknowledge the significant influence of microbial community composition and abundance on decomposition mechanisms, I am uncertain whether this study offers a sufficiently detailed mechanistic understanding. Please see my detailed comments below for further clarification on this.

The results and discussion sections are clear and well-organized. The other sections could benefit from revision. The abstract and conclusions particularly lack essential discussions that could enhance their impact. In addition, I suggest reorganizing certain paragraphs in the introduction and methods sections to enhance the overall flow. I also recommend supplementing the introduction and discussion sections with relevant citations to bolster the scholarly foundation of the manuscript. Detailed comments are provided below for your consideration.

Detailed comments:
- Line 21: Please revisit this sentence, considering my comment above on the limitations of this study. Also, what is a range of realistic proportions of flower:leaf litter?
- Line 26-27: 'while leaf litter had a higher concentration of Ca, Mg, Na' – What are the implications of this result? How is this important to understand the characteristics of leaf litter and their role on litter-mixing effects?
- Consider mentioning the labile and recalcitrant nature of flower and leaf litter in the abstract.
- Line 27: Consider incorporating this statement "To our knowledge, …" earlier in the abstract. Be concise. (e.g., Line 18, 'for the first time')
- Line 33: Include references across terrestrial and aquatic ecosystems
- Line 48: add relevant reference(s)
- Line 89-105: Consider moving this background information earlier for better flow.
- Consider moving section 2.4 earlier in the methods, as this section pertains to characterizing the initial chemical compositions of flower and leaf litter. I would prefer to read this before the experimental setup.
- Also, consider moving LMPexp (eq. 3) and RME (eq. 4) to section 2.4 and reserve section 2.5 for statistical analysis.
- Line 203-204: Briefly explain the analytical methods/protocols used to determine K, Ca, Mg, and Na contents
- Line 215: three 3 times → three times
- Line 222-225: Clarify this for better understanding.

- Line 278: consistent to → consistent with
- Line 365: Specify what the 'second prediction' was.
- Line 440: Revisit this sentence "However, ……" for clarification.
- Line 479-491: Elaborate on the effects of soil invertebrates and their importance in litter-mixing effects in decomposition. Briefly mention how they affect LMEs. Include references for both caveats mentioned.
- Line 539: the importance litter from … → the importance of litter from …
- Strengthen the conclusions by integrating key points made in the results and discussion sections. I recommend that the authors bolster the conclusions by reiterating important implications of the study, such as those outlined in line 68-71 and line 402-404.

---

## Author Response (AR1)

**Associate Editor' comments**

Dear Authors,

Thank you for providing detailed responses to the comments and suggestions offered by the two reviewers.

Both reviewers recognized the significance of your work and agreed that the manuscript should be conditionally accepted after some minor revisions. I also agree with the reviewers that your manuscript sheds new light on the old topic of litter decomposition by examining flower-enhanced litter mixing effects in both terrestrial and aquatic ecosystems. Based on the positive evaluations of the two reviewers and my careful perusal, I recommend conditional acceptance of the manuscript with the assumption that some moderate revision will be needed to bring the manuscript up to an acceptable level and clarity. Please also refer to my following comments on your responses:

**- Responses that require further clarification**

Response: Indeed, the presence of macro-fauna was limited in our experiment, and the litter decomposition was mediated by micro- and meso-fauna (as highlighted in the discussion; see (lines 494-508). The presence of fauna in experiments has been shown to increase litter decomposition rate and also intensify the LMEs, accentuating the occurrence and magnitude of synergistic effects on decomposition (Njoroge et al. 2023). Our results showed very consistent synergistic LMEs even in the absence of fauna. For these reasons, although we may agree that the absence of macro-fauna could have quantitatively altered the litter decomposition rates, we refrain from believing that the absence of macro-fauna could have altered our results qualitatively by modifying the occurrence and direction of litter-mixing effects we have observed.

--> The observed consistent synergistic LMEs in the absence of fauna cannot exclude the stimulating role of fauna. Please articulate this issue in Discussion.

**Response:** Thank you for your suggestion. We added more details on how the exclusion of fauna could limit a realistic quantification of LMEs of flower and leaf litter interaction in the discussion as, "Secondly, the absence of macrofauna in our experiment could limit

an accurate estimation of LMEs through flower and leaf litter interaction, since the presence of macro-fauna has been repeatedly shown to be an important factor in determining the occurrence and magnitude of synergistic LMEs on decomposition through litter fragmentation and decomposers complementary use of litter resources (Hättenschwiler and Gasser, 2005; Njoroge et al., 2022, 2023). Therefore, in future studies the inclusion of macrofauna could be important to quantify the real impact of flower and leaf litter interaction in nutrient dynamics in ecosystems" (Lines 504-509).

References:

Hättenschwiler, S, and Gasser, P.: Soil animals alter plant litter diversity effects on decomposition. Proc Natl Acad Sci, 102, 1519-1524, https://doi.org/10.1073/pnas.0404977102, 2005.

Njoroge, D. M., Chen, S.-C., Zuo, J., Dossa, G. G. O., and Cornelissen, J. H. C.: Soil fauna accelerate litter mixture decomposition globally, especially in dry environments, J. Ecol., 110, 659–672, https://doi.org/https://doi.org/10.1111/1365-2745.13829, 2022.

Njoroge, D. M., Dossa, G. G. O., Ye, L., Lin, X., Schaefer, D., Tomlinson, K., Zuo, J., and Cornelissen, J. H. C.: Fauna access outweighs litter mixture effect during leaf litter decomposition, Sci. Total Environ., 860, 160190, https://doi.org/https://doi.org/10.1016/j.scitotenv.2022.160190, 2023.

Response: We used paper filters to quantify the small particulate organic matter associated with flower litter, which fragmented more easily (personal observation). This problem did not occur with leaf litter, which disintegrated into larger particles at the end of the experiment. However, to maintain the same weighing procedure between the two litter types, we filtered them through paper filters and then quantified their mass loss separately.

--> This (the next one as another) is another example of unclarified responses. Please make sure how you would clarify this and other unarticulated responses in the revised manuscript.

**Response:** The question raised by the reviewer can also be a question to the readers. To make clear, we added the information above in the methods section, "We used paper filters to quantify the small particulate organic matter associated with flower litter, which

fragmented more easily (personal observation). This problem did not occur with leaf litter, which disintegrated into larger particles at the end of the experiment. However, to maintain the same weighing procedure between the two litter types, we filtered them through paper filters and then quantified their mass loss separately" (Lines 209-213).

Answer: Thank you for your question. Indeed, we assumed that small litter particles were dominant or even entirely from flower litter. First, the leaf litter was still in large and identifiable sizes, at the end of the experiment. The differences in paper filter mass before and after the filtration of leaf litter were minimal (t-test= 0.95; p=0.78). Thus, there was no overestimation of flower litter mass due to the fine leaf particles. Secondly, if leaf litter made a high contribution to the proportion of the fine litter particles, we should observe an increasing "contamination" and underestimation of the flower litter decomposition with the increase of leaf litter proportion in the mixture, resulting in antagonistic effects on flower litter, a result contrary to that we have observed. Thus, our results suggest that the possible experimental artifact cited had no relevant effect on our results.

**Response:** To make it clear to the readers, we added the information about the t-test in the methods section, "To verify if the paper filter could retain fine particles only for flower litter, we compared the paper filter mass before and after filtration of leaf litter to guarantee that there was no overestimation of flower litter mass due to fine leaf litter particles retained in paper filter, though the comparison of paper filter mass before and after leaf litter filtration of leaf litter in monoculture treatments (t-test= 0.95; p=0.78)" (Lines 219-222).

- Regarding your response to the second reviewer's comment on "range of realistic proportions of flower:leaf litter", please also take into consideration temporal variations in the flower contribution to the total litter to discuss the implications of your incubation results for assessing litter mixing effects under the "real-world" conditions. Although it would be difficult to find "accurate" information, a quick literature survey could help you provide rough estimates of flower-leaf ratios and their seasonal variations. If available, any information on the quantitative importance and chemical composition of flower litter (in any similar or different systems) would be very helpful. This type of empirical information is critical given your goal ("To obtain realistic results and a mechanistic

understanding"). Based on the empirical information, please discuss the role of flower-enhanced decomposition under actual field conditions (if possible, also for ecosystems in other climate zones).

**Response:** We improved our motivations for carrying out the experiment in a range of flower:leaf biomass proportions by adding more information from other studies about the temporal variation of flower litter into litter layer in the methods, "It is important to note that the flower:leaf biomass proportion in the litter layer can significantly vary in nature across space and time. This variation can be attributed to several factors such as plant species identity, individual size, timing and magnitude of flowering phenology, and distance from the plant originating the litter (Uriarte et al. 2015; Buonaiuto and Wolkovich, 2021). However, despite the significance of this information, the literature still lacks data on flower:leaf biomass proportion for the majority of species, including the species used in our study. Although, a recent study looked at the amount of flower and leaf litter biomass for several species. The study found that despite leaf litter is generally more common than flower litter on an annual basis, the amount of flower and leaf litter varies significantly throughout the year. As a result, the proportion of flower:leaf biomass in the litter layer can vary greatly for different species. On average, flower litter contributes around 25% of the leaf litter on an annual basis, but this can range from 5% to 45% (Hill et al. 2022). And in some cases, during the blooming season, the amount of flower litter can even exceed the amount of leaf litter (Wang et al. 2016). Therefore, to encompass the unknown and possibly extensive variability in flower:leaf biomass proportion that may occur for *T. aurea* in nature, we assembled mixtures of flower and leaf litter along a gradient encompassing nine different flower:leaf biomass proportion" (Lines 167-179).

References:

Buonaiuto, D. M. and Wolkovich, E. M.: Differences between flower and leaf phenological responses to environmental variation drive shifts in spring phenological sequences of temperate woody plants.J. Ecol., 109, 2922–2933. https://doi.org/10.1111/1365-2745.13708, 2021.

Hill, S. K., Hale, R. L., Grinath, J. B., Folk, B. T., Nielson, R., and Reinhardt, K.: Looking beyond leaves: variation in nutrient leaching potential of seasonal litterfall among

different species within an urban forest. Urban Ecosyst 25, 1097–1109, https://doi.org/10.1007/s11252-022-01217-8, 2022.

Uriarte, M., Turner, B. L., Thompson, J. and Zimmerman, J. K.: Linking spatial patterns of leaf litterfall and soil nutrients in a tropical forest: a neighborhood approach. Ecol. Appl., 25: 2022-2034. https://doi.org/10.1890/15-0112.1, 2015.

Wang, J., Xu, B., Wu, Y., Gao, J., and Shi, F.: Flower litters of alpine plants affect soil nitrogen and phosphorus rapidly in the eastern Tibetan Plateau, Biogeosciences, 13, 5619–5631, https://doi.org/10.5194/bg-13-5619-2016, 2016.

- Your response to the lability of flower litter ("Flower litter had a higher concentration of nutrients important to decomposition, such as N, P, K, and water-holding capacity representing a labile litter"): Anybody who has some research experience in organic matter decomposability would specify "labile organic components" as the primary lability marker. Please base your description here (and also in Introduction and Discussion) on some primary (measured) or secondary (literature) information on the litter composition that would be more relevant for assessing the lability of litter.

**Response:** We assumed that flower litter was more labile due to the higher nutrient concentration, such as N, P, and K, but also due to the lower constitutive defenses against herbivory and structural tissues, as we mentioned in the introduction (Lines 76-84) and discussion (Lines 401-417). To make clear in the abstract, we rephrased the sentence as "Flower litter had a higher concentration of labile C-compounds, N, P, and K and lower lignin concentrations representing a labile litter; while leaf litter had a higher concentration of lignin, Ca, Mg, and Na, representing a refractory litter." (Lines 23-26).

- In terms of creating realistic incubation conditions, incubating litter in tap water for three months has a clear limitation in simulating any lotic (and even lentic) system. Please articulate which type of freshwater system you wanted to simulate and discuss the limitations of your incubation results in applying to any "real-world" aquatic systems.

**Response:** We described better which kind of aquatic environments we wanted to simulate adding the possible caveats in our experimental design, as "...in the aquatic experiment, we simulated stillwater environmental conditions typically observed in lentic

systems, such as temporary pools along the channel of intermittent rivers, small ponds, phytotelmata, and so forth. These environments are generally nutrient-poor and result in the prolonged confinement of water and litter (Migliorini et al., 2018; Bonada et al, 2020), potentially affecting the generalizability of our results to other aquatic systems. However, our incubation method may not fully replicate real-world conditions, especially within lotic ecosystems. Therefore, future studies should assess the occurrence, magnitude, and direction of LMEs resulting from the interaction between flower and leaf litter in lotic systems. In these systems, there is a long tradition of studies evaluating the decomposition of detritus from riparian vegetation, yet the importance of the interaction between leaf and flower litter in decomposition is often overlooked (but see Rezende et al. 2017)" (Lines 509-517).

Furthermore, in the methods section, we added more details concerning how we used dechlorinated tap water and added a water inoculum to guarantee the colonization of microorganisms as "Dechlorinated tap water was used to fill the aquatic experimental microcosms and the water inoculum from the oligotrophic Carcará Lake (6°3´40"S, 35°9´28"W) was added to allow the colonization of microorganisms" (Lines 191-193).

References:

Bonada, N., Cañedo-Argüelles, M., Gallart., F. von Schiller, D., Fortuño, P., Latron, J., Llorens, P., Múrria, C., Soria, M., Vinyoles, D., and Cid, N.: Conservation and management of isolated pools in temporary rivers. Water, 12, 2870, https://doi.org/10.3390/w12102870, 2020.

Migliorini, G. H., Srivastava, D. S., and Romero, G. Q.: Leaf litter traits drive community structure and functioning in a natural aquatic microcosm. Freshwater Biology 63.4, 341-352, https://doi.org/10.1111/fwb.13072, 2018.

Rezende, R., R.S. Correia, P., Goncalves Jr, J., and Santos, A.: Organic matter dynamics in a savanna transition riparian zone: Input of plant reproductive parts increases leaf breakdown process, https://doi.org/10.4081/jlimnol.2017.1601, 2017.

When you have completed revising the manuscript, I would like to ask you to make all the changes easily identifiable in a marked-up manuscript based on your point-by-point

responses to the reviewer comments. If possible, please specify the line numbers of the revised parts in your final responses accompanying the revised manuscript

**Response:** All changes in the revised manuscript were highlighted in yellow, including the reviewers' and associate editor' comments. Below I updated the answer to the reviewers' comments.

Sincerely,

Ji-Hyung Park

Associate Editor, Biogeosciences

**#Reviewer 1**

**General comments:**

**The manuscript reports a very interesting study on litter decomposition on aquatic and terrestrial ecosystems, and how it is affected by mixing qualitatively distinct litter. These topics are not particularly novel, but the authors test if and how the interaction of distinct proportions of plant components (flowers and leaves) from a single species (*Tabebuia aurea*) affect the occurrence, direction, and magnitude of litter mixing effects on decomposition, which is a novel aspect of the study. By doing so, the study demonstrates that litter mixing effects on decomposition can also occur intraspecifically via the interaction of distinct plant components, and calls attention to the importance of mixing effects on decomposition even in low-diverse systems through the interactions of litter from different plant organs.**

**I find the research questions interesting, but had some difficulty in understanding the methods used to evaluate litter decomposition on aquatic conditions, which require further clarification. Additionally, the calculations (or simply the formulae) considered for estimating litter mixing effects on decomposition need to be reviewed.**

**Overall, results are clearly presented and discussions are very straightforward. Writing is good, but I recommend you have some check on linguistic issues (I am not a native speaker myself, hence I understand the challenges involved).**

**Overall, I think it will make a valuable contribution to Biogeosciences after some minor revisions.**

**For detailed comments, please see the feedback provided below.**

**Response:** Dear reviewer, we are delighted by your positive impression of our work and immensely grateful for your constructive feedback. We are confident that our work has greatly improved thanks to your critiques, suggestions, and comments. Below, we address each of your comments.

**Specific comments:**

**Line 34: The work fails to cite classic or pivotal researches that set up their study. Among important works, I would suggest: Wagener et al. 1998 (Rivers and Soils: Parallels in Carbon and Nutrient Processing); Cebrian & Lartigue, 2004 (Patterns of herbivory and decomposition in aquatic and terrestrial ecosystems); Tiegs et al., 2019 (Global patterns and drivers of ecosystem functioning in rivers and riparian zones).**

**Response:** We thank the reviewer for pointing this out, we replaced the reference, adding Cebrian and Lartigue (2004) and Tiegs *et al.* 2019 (Lines 32-33), and updated the references section (Lines 650-651; 860-872).

**Line 141: Lidman et al. (2017) is more concerned about plant community composition than environmental effects on decomposition. Please, review this reference… or simply remove the sentence, since you inform in the next sentence the duration of experiment in each environmental condition tested.**

**Response:** We thank the reviewer for pointing this out, we removed the sentence and maintained the information of experiment duration for aquatic and terrestrial ecosystems. Thus, the text excerpt is: "The duration of the terrestrial and aquatic experiments was standardized by the time required for approximately 50% of the more labile litter (i.e. flowers) to be decomposed in each environment. The aquatic experiment lasted for 3 months, while the terrestrial experiment lasted for 7 months." (Lines 158-161).

**Line 143 – 159: The presence of fauna have a considerable effect on litter decomposition, what could imbalance results within and among treatments and ecosystems. Were these microcosms ever censused for fauna access? How do you dealt with that?**

**Response:** Indeed, the presence of macro-fauna was limited in our experiment, and the litter decomposition was mediated by micro- and meso-fauna (as highlighted in the discussion; see (lines 504-509). The presence of fauna in experiments has been shown to increase litter decomposition rate and also intensify the LMEs, accentuating the occurrence and magnitude of synergistic effects on decomposition (Njoroge et al. 2023).

Our results showed very consistent synergistic LMEs even in the absence of fauna. For these reasons, although we may agree that the absence of macro-fauna could have quantitatively altered the litter decomposition rates, we refrain from believing that the absence of macro-fauna could have altered our results qualitatively by modifying the occurrence and direction of litter-mixing effects we have observed.

Njoroge, D. M., Dossa, G. G. O., Ye, L., Lin, X., Schaefer, D., Tomlinson, K., Zuo, J., and Cornelissen, J. H. C. (2023). Fauna access outweighs litter mixture effect during leaf litter decomposition. The Science of the total environment, 860, 160190. https://doi.org/10.1016/j.scitotenv.2022.160190

**Line 169: Was fauna access allowed to the aquatic microcosms also?**

**Response:** It was not. Both experiments were carried out under laboratory conditions excluding macrofauna and the decomposition was mainly driven by microbes.

**Lines 179 - 184: The description of the methodology for determination of final mass in mixtures at aquatic microcosms is not clear to me. Why using paper filter to dry leaf litter if you could separate them from flower litter? Was dissolved + particulate material also filtered on paper filters containing leaf litter, as you did for monocultures? If not, why don't simply weighting it on aluminum trays after separation?**

**Response:** We used paper filters to quantify the small particulate organic matter associated with flower litter, which fragmented more easily (personal observation). This problem did not occur with leaf litter, which disintegrated into larger particles at the end of the experiment. However, to maintain the same weighing procedure between the two litter types, we filtered them through paper filters and then quantified their mass loss separately. To enhance the clarity of this part for the reader, we've rephrased the sentence in lines 209-213 of the main text with the above justifications.

**In Lines 182 - 183, you mentioned you "measured the remaining mass of each litter type in its respective microcosm, allowing us to estimate the decomposition rate of each litter type individually, even in mixtures." Was the final flower litter mass determined as this "remaining mass" in paper filters after separation from leaf litter? However, as you mentioned for monocultures (Line 179), the remaining water contained particulate material, what I presume may be also common for mixtures. Was the final flower litter mass not overestimated due to fine leaf particles retained in the paper filters? Do you have any information about how much fine particles accounted for the final leaf litter mass in monocultures? Maybe it could give an idea about such overestimation, if it is the case.**

**Answer:**. Indeed, we assumed that small litter particles were dominant or even entirely from flower litter. First, the leaf litter was still in large and identifiable sizes, at the end of the experiment. The differences in paper filter mass before and after the filtration of leaf litter were minimal (t-test= 0.95; p=0.78). To enhance the clarity of this part for the reader, we've rephrased the sentence in lines 219-222 of the main text with the above justifications

**Line 199: How many replicates did you consider for chemical analyses?**

**Response:** We thank the reviewer for pointing this out, we did three replicates to measure each litter trait. We added this information in the section *Estimation of flower and leaf litter functional traits:* " Each functional trait had three replicates" (Line 133).

**Lines 213 - 216: It is not clear why did you measure WHC and leaching along the gradient of variation of flower:leaf litter. Where are these results presented and discussed also?**

**Response:** We thank the reviewer for pointing this out. We have corrected this information in the methods: "To evaluate physical traits, we assessed the water-holding capacity (WHC) and leaching of flower and leaf litter, which are considered crucial factors in determining litter decomposition (Makkonen et al., 2013). To evaluate WHC, we used dried flower and leaf litter and moistened the replicates with 50 ml of water (the same volume used to irrigate the terrestrial experiment) two hours before the

measurements, based on Makkonen et al. (2013). For the leaching measurement, we stimulated the loss of hydrolysable water compounds, which is the main form of mass loss in the initial stage of decomposition, based on Pérez-Harguindeguy et al. (2013). For both structural traits, the material was dried in an oven at 60 °C for 72 h before and after the measurements. " (Lines 143-149).

Initially, we measured WHC and leaching considering treatments with different flower-to-leaf ratios to test for the possibility of different litter mixture proportions affecting WHC and leaching. However, we did not observe changes in WHC and leaching compared to the mixture and individual litter types. Therefore, for simplicity of the results, we presented only the data for the individual litter types, but we forgot to update this information in the submitted version of the manuscript.

**Line 221: How many replicates were considered for evaluating litter traits differences among plant organs?**

**Response:** We thank the reviewer for pointing this out. We used three replicates, for each litter trait, to evaluate the differences in the litter traits among organs. We added the number of replicates for each trait in the methods: "Each functional trait had three replicates." (Line 133).

**Lines 232 – 238: The method you described is equivalent to an Analysis of Covariance (ANCOVA), and its calculations follow a method presented in Chapter 18 of J Zar, Biostatistical Analysis, 2nd edition, Prentice-Hall, 1984. Please, cite if it is the case.**

**Response:** We thank the reviewer for pointing this out. We added the reference suggested and explicitly said that the method used is equivalent to ANCOVA, as "This method is equivalent to an Analysis of Covariance (ANCOVA) (Zar, 1984)" (Lines 262-263) and in the references section (Lines 894-895).

**Line 239: Why did you choose to use an unpaired t-test (instead of a paired)? Once you have LMR along flower:leaf litter gradient, shouldn't you use a paired t-test among flower:leaf biomass levels?**

**Response:** We thank the reviewer for the question. We ran t-tests just in case we did not observe the interactive effects of the proportion of litter mixture on the litter-mixing effects respective to flower litter, leaf litter, and mixture. In this case, the t-test was used to test the significance of the main effect of the mixture of the two litter types on litter-mixing effects, regardless of proportion (grand mean effect). In the scenario mentioned, the samples are independent, which is why the unpaired test was chosen.

**Line 244 - 245: It is not clear what your objectives in this sentence are. What do you mean by 'to understand the magnitude'? Did you mean 'individual litter types in mixtures, as well as the whole mixture'? I think it would make it clearer if you keep only the ideas concerning the RME estimation in the sentence, not introducing your intention to associate it to flower litter biomass proportion (since you already do it in the end of the paragraph).**

**Response:** We thank the reviewer for pointing this out. In fact, this sentence was confusing. So, we removed the motivation about equation 4 and just kept the description: "Then, we calculated the relative mixture effect (RME) in each litter type for each ecosystem (Barantal et al., 2011) using Eq. (4):" (Lines 239-240).

**Line 246: Equation 4 should be revised. According to Barantal et al. (2011), RME (%) = ((LMRobs - LMRexp) / LMRexp) × 100. Please check if it is necessary to correct only the formulae or the calculations also.**

**Response:** We thank the reviewer for pointing this out. In fact, there was a misspelling mistake in the equation, but we calculated the RME based on Barantal *et al.* (2011). So, we corrected the equation according to Barantal *et al.* 2011, RME (%) = (($LMR_{obs}$ - $LMR_{exp}$) / $LMR_{exp}$) × 100 (Equation 4, Line 241).

**Lines 312 – 314: The absolute difference among leaf LMR in mixtures and in monocultures is 3.2%. However, the value provided corresponds to 4.5%. How was this calculated? Is it mentioned in the methods section? Also check Lines 318 and 325.**

**Response:** We thank the reviewer for pointing this out. We corrected the percentage of differences between the grand means of LMR in monoculture and mixture (Lines 327, 332, and 339).

**Line 348: 'Fig. 4a' -> Fig 4b. By the way, the right panel indicating the RME for individual flower and leaf litter in mixtures, on aquatic experiment is not represented on Figure 4b**

**Response:** We thank the reviewer for pointing this out. We changed the indication for Fig. 4b: "Furthermore, average values of RMEs for flower (7%) and leaf litter (6%) were not significantly different from each other (Fig. 4b right panel; $t=0.34$; $p=0.74$ unpaired t-test)"; and updated Fig. 4 with the right panel (Lines 365-368).

**Line 376: Lin et al., 2019 are not concerned about the effects of litter mixing from different plant organs for the occurrence of LMEs on decomposition**

**Response:** We thank the reviewer for pointing this out. We replaced the Lin *et al.* (2019) with the following references (Line 395):

de Paz, M., Gobbi, M. E., and Raffaele, E.: Fallen fruits stimulate decomposition of leaf litter of dominant species in NW Patagonia shrublands, Plant Soil, 425, 433–440, https://doi.org/10.1007/s11104-018-3590-0, 2018

Hou, S.-L. and Lü, X.-T.: Mixing effects of litter decomposition at plant organ and species levels in a temperate grassland, Plant Soil, 459, 387–396, https://doi.org/10.1007/s11104-020-04773-0, 2021.

**Line 410: In fact, you rejected your third hypothesis, since there were not significant differences among the magnitudes of flower on leaf litter decomposition or vice-versa.**

**Response:** We thank the reviewer for pointing this out. We changed the sentence informing that we rejected our third hypothesis: "However, our findings rejected our third prediction" (Line 429).

**Technical corrections:**

**Line 21: we manipulated various scenarios of flower-to-leaf litter biomass proportion…**

**Response:** We thank the reviewer for pointing this out. We changed the sentence as suggested (Lines 19-20).

**Line 24: … and at higher flower:leaf litter biomass proportions**

**Response:** We thank the reviewer for pointing this out. We changed the sentence as suggested: "Litter-mixing effects were stronger in the terrestrial environment and at higher flower:leaf litter biomass proportions." (Line 22).

**Line 24: 'Our results indicate that synergistic results are' -> Our results indicate that synergistic outcomes are…**

**Response:** We thank the reviewer for pointing this out. We changed the sentence as suggested: "Our results indicate that synergistic outcomes are mainly associated with complementary effects" (Lines 22-23).

**Line 28: These results shed light...**

**Response:** We thank the reviewer for pointing this out. We changed the sentence as suggested: "These results shed light on the secondary consequences of flower litter on

decomposition, suggesting that species with high reproductive investment in flower biomass may play an important role in the nutrient and carbon recycling of diverse plant communities, exerting a pivotal role in biogeochemical dynamics" (Line 26).

**Line 48: "The chemical compounds and physical structures of plant litter in ecosystems are highly diverse and heterogeneous, leading to distinct litter quality" – Reference?**

**Response:** We thank the reviewer for pointing this out. We added the following references to support our ideas in the sentence (Lines 47-48):

Freschet, G. T., Cornelissen, J. H. C., van Logtestijn, R. S. P., and Aerts, R.: Substantial nutrient resorption from leaves, stems and roots in a subarctic flora: what is the link with other resource economics traits?, New Phytol., 186, 879–889, https://doi.org/https://doi.org/10.1111/j.1469-8137.2010.03228.x, 2010.

Freschet, G. T., Cornwell, W. K., Wardle, D. A., Elumeeva, T. G., Liu, W., Jackson, B. G., Onipchenko, V. G., Soudzilovskaia, N. A., Tao, J., and Cornelissen, J. H. C.: Linking litter decomposition of above- and below-ground organs to plant–soil feedbacks worldwide, J. Ecol., 101, 943–952, https://doi.org/https://doi.org/10.1111/1365-2745.12092, 2013.

Olson, M. E. and Pittermann, J.: Cheap and attractive: water relations and floral adaptation, New Phytol., 223, 8–10, https://doi.org/https://doi.org/10.1111/nph.15839, 2019.

Schmitt, L. and Perfecto, I.: Who gives a flux? Synchronous flowering of Coffea arabica accelerates leaf litter decomposition, Ecosphere, 11, e03186, https://doi.org/https://doi.org/10.1002/ecs2.3186, 2020.

**Line 74: See also: Wang et al. 2022 (Litter diversity accelerates labile carbon but slows recalcitrant carbon decomposition)**

**Response:** We thank the reviewer for pointing this out. We added the suggested reference (Lines 89-90).

**Line 87: occurrence, direction and magnitude**

**Response:** We thank the reviewer for pointing this out. We added the word "direction" to the sentence as suggested: "To better understand the possible mechanisms underlying LMEs on decomposition, we evaluated the occurrence, magnitude, and direction of LMEs on each litter type individually and on the whole litter mixture." (Line 102).

**Line 96: Insert "," after 'growth-rate hypothesis'**

**Response:** We thank the reviewer. We inserted the "," to the sentence as suggested (Lines 77-78).

**Line 162: 'varied' -> adjusted**

**Response:** We thank the reviewer for pointing this out. We replaced the word in the sentence as suggested: "However, as the final litter biomass added to the microcosms differed between mixtures and their respective monocultures, as well as throughout the monocultures, the volume of water in each microcosm was adjusted to maintain a final litter concentration of 3 g/L across all treatments" (Line 195).

**Line 173: For how long and at which temperature was litter dried?**

**Response:** We thank the reviewer for pointing this out. The litter was dried at 60 °C for 72h. We added this information in the sentence: "For mixtures, the remaining flower and leaf litter were visually identified and separated and subsequently dried at 60 °C for 72 h and weighed to estimate litter mass loss." (Line 205).

**Line 176: 'prohibited the same procedure' -> limited a similar procedure**

**Response:** We thank the reviewer. We changed the sentence as suggested: "In the aquatic experiment, flower litter fragmentation limited a similar procedure" (Line 208).

**Line 182: 'filter paper' -> paper filter**

**Response:** We thank the reviewer for pointing this out. We changed the term as suggested (Line 217).

**Line 196: 'Estimative' -> Estimation**

**Response:** We thank the reviewer. We changed the word as suggested (Line 130).

**Line 214: 'since only flower litter' -> ranging from pure flower litter**

**Response:** We thank the reviewer for pointing this out, but we deleted this part of the text based on this suggestion "Lines 213 - 216: It is not clear why did you measure WHC and leaching along the gradient of variation of flower:leaf litter. Where are these results presented and discussed also?"

**Line 222: What does the expression 'or more' refer to in this sentence?**

**Response:** We thank the reviewer for pointing this out. In fact, the term "or more" is not suitable here, since we only had two litter types. For this, we deleted this term from the sentence (Lines 239-240).

**Line 252: I think you need a word 'respectively' after 'monoculture', associated to the expression 'positive and negative values'**

**Response:** We thank the reviewer. You are right, we added the word "respectively" to the sentence: "For RMEs, positive and negative values indicate that litter decomposes faster and slower in mixtures than in its respective monoculture, respectively" (Lines 246-247).

**Lines 252 – 254: It is a little bit hard to understand the spelling and the meaning of this sentence. Please check and rephrase it.**

**Response:** We thank the reviewer for pointing this out. We rephrased the sentence to: "To assess whether the RME for each type of litter and the mixture is a function of their respective biomass in the mixture, we utilized linear regressions for both aquatic and terrestrial experiments" (Lines 267-268).

**Line 277: 'was' -> being**

**Response:** We thank the reviewer for pointing this out. We changed the word as suggested (Line 291).

**Line 278: 'This pattern was consistent to the aquatic experiment, the values… ' -> This pattern was consistent with that observed in the aquatic experiment, and the values…**

**Response:** We thank the reviewer for pointing this out. We changed the sentence as suggested: "This pattern was consistent with that observed in the aquatic experiment, and the values of leaf and flower litter decomposing in monoculture were significantly different from each other" (Lines 292-294).

**Line 283: The word 'mixed' seems a bit redundant in this sentence**

**Response:** We thank the reviewer. We deleted the word "mixed" from the subhead (Line 297).

**Line 293: 'and LMEs on…' seems out of place. Also, you do not present LME results in Figure 2b**

**Response:** We thank the reviewer for pointing this out. We deleted the information about LMEs (Line 303).

**Line 310: The word 'mixed' seems a bit redundant in this sentence**

**Response:** We thank the reviewer. We deleted the word "mixed" from the subhead (Line 324).

**Lines 322 – 323: What do you mean by 'the LMEs on the decomposition of the whole mixture were significant on average'?**

**Response:** We thank the reviewer. We meant that the litter mass remaining observed differed from the litter mass remaining expected. We rephrased the sentence completing the information: "Nevertheless, the LMEs on the decomposition of the whole mixture were significant (Fig. 3c, right panel; t=2.3; p=0.03, unpaired t-test), as the average observed LMR for the litter mixture (57.5%) was significantly lower than its expected value (61.5%), calculated from the decomposition of both litter types alone, indicating that litter mixing had, on average, a stimulating effect of 4% on the decomposition of the whole litter mixture." (Lines 336-339).

**Lines 340 – 341: 'varied significantly and negatively as a function of flower:leaf litter proportion (Fig 4a)' -> positively… increasing flower:leaf litter improves RME**

**Response:** We thank the reviewer. We changed the sentence to: " In the terrestrial experiment, RME values for flower, leaf, and both litter types combined, varied significantly and positively as a function of flower:leaf litter proportion (Fig 4a)" (Lines 356-358).

**Line 347: … 'for whole mixture' in the aquatic experiment**

**Response:** We thank the reviewer. We changed the sentence as suggested: "Contrary to what was observed in the terrestrial experiment, we did not observe significant effects of flower:leaf litter biomass proportion on the variation of RME values for leaf and flower litter as well as for whole mixture in the aquatic experiment (Fig. 4b)" (Lines 363-365).

**Line 358: 'LMEs of mixing flower and leaf litter' -> LMEs of flower and leaf litter mixture**

**Response:** We thank the reviewer. We changed the sentence as suggested: "Our study is the first to assess the LMEs of flower and leaf litter mixture on decomposition across terrestrial and aquatic ecosystems" (Line 376).

**Line 365: 'flower and leaf mixing' -> flower and leaf litter mixture**

**Response:** We thank the reviewer. We changed the sentence as suggested: "...since the occurrence of LMEs of the flower and leaf litter mixture were consistent in terrestrial and aquatic environments. " (Line 384).

**Line 495: 'leaves' -> leaf**

**Response:** We thank the reviewer. We changed the word "leaves" to "leaf" as suggested (Line 520).

**Line 540: 'have evidenced about the role' -> 'evidence about' ou 'have evidenced the role'**

**Response:** We thank the reviewer. We changed the sentence "have evidenced about the role" to "have evidenced the role" (Line 569).

**Reviewer #2**

This manuscript presents the effects of varied proportions of leaf and flower litter on litter mixing effects in terrestrial and aquatic ecosystems. The study design and findings are interesting, particularly the investigation on intra-specific interactions on decomposition, which I found to be a novel contribution to the field. I believe the study presents valuable insights and will inspire further research in this area. Given the important role of litter-mixing effects in nutrient and carbon dynamics, the manuscript fits well within the scope of BG.

**However, I have a concern regarding the limitations of the study. While the authors acknowledge the significant influence of microbial community composition and abundance on decomposition mechanisms, I am uncertain whether this study offers a sufficiently detailed mechanistic understanding. Please see my detailed comments below for further clarification on this.**

**The results and discussion sections are clear and well-organized. The other sections could benefit from revision. The abstract and conclusions particularly lack essential discussions that could enhance their impact. In addition, I suggest reorganizing certain paragraphs in the introduction and methods sections to enhance the overall flow. I also recommend supplementing the introduction and discussion sections with relevant citations to bolster the scholarly foundation of the manuscript. Detailed comments are provided below for your consideration.**

**Response:** We are sincerely grateful for your constructive comments and critiques, and we are pleased by your overall positive impression of our work. Below, we have carefully considered each criticism and comment, hoping to address your questions and concerns regarding the understanding of potential mechanisms at play in our study.

**Detailed comments:**

**• Line 21: Please revisit this sentence, considering my comment above on the limitations of this study. Also, what is a range of realistic proportions of flower:leaf litter?**

**Response:** We thank the reviewer for the comments. We agree that maybe we did not have a full mechanistic understanding on what factors mediated the flower and leaf litter interaction, since for this, we would need to quantify a series of microbial-mediated processes, such as nutrient transfer, to identify which mechanisms resulted in the observed LMEs. So, we rephrase the sentence to avoid a misinterpretation to: "To obtain realistic results, we manipulated various scenarios of flower-to-leaf litter biomass proportion and measured 13 functional traits, respectively" (Lines 19-20).

However, although we cannot ascertain which microbial-mediated mechanisms were most important in determining the observed LMEs, we can safely assert that their effects resulted in mechanisms of complementarity since each litter type decomposed

faster in the litter mixture than when it was alone (a phenomenon known as transgressive overyielding, which has been recognized as unequivocal evidence of complementarity effects ever since the seminal papers that introduced the first metrics for assessing how biodiversity mechanisms influence ecosystem processes; see Loreau 1988; Hector 1988; Loreau & Hector 2001). Therefore, our results have implications for the importance of LMEs at the intra-specific level, concerning the flower and leaf litter interactions, especially regarding the gradient of flowers-to-leaves litter biomass proportions in mixtures. We added a section in the discussion that advocates for a more detailed explanation of the mechanisms in future studies and the implications of flower and leaf litter interactions in natural ecosystems (Lines 499-517).

Now, in response to the reviewer's inquiry regarding the real-world scenarios concerning the proportions of flower-to-leaf litter biomass in nature, we are unaware of any available information regarding the input rates of flower fall biomass or the standing stock of flower biomass in the litter layer for the species we studied. This impairs us from providing accurate information regarding the flower-to-leaf litter biomass proportion observed in nature for the species *T. aurea*. However, one of our motivations for carrying out the experiment was to understand how the natural variations in the flower-to-leaf litter biomass proportion around the individual and between individuals of *T. aurea* can alter the LMEs, as mentioned in the methods (Lines 167-179). Therefore, in addition to the primary motivations behind our experimental design, which aimed to test the effects of mixing different proportions of ROM (leaf) and LOM (flower) on decomposition (as outlined in the introduction, Lines 99-101), our intention to manipulate a broad gradient of flower-to-leaf litter mixture proportions was also due to the lack of information about a real average ratio of the mixture of these two litter types in nature (also mentioned in the methods, Lines 167-179). Thus, by testing the LMEs across a gradient of flower-to-leaf litter mixture proportions, we gain insights into whether and how the flower-to-leaf litter mixture proportion may affect the LMEs.

Furthermore, our results indicate that the direction and magnitude of LMEs varied as a function of flower:leaf litter proportion, which gave us confidence in our choice to use a gradient of flower:leaf litter proportion rather than just an average proportion value that better captured the effects of the interaction between flower and leaf litter on LMEs, particularly in the terrestrial ecosystem since proportion was unimportant in the aquatic experiment. Although to accurately understand the effects of flower litter on LMEs, we

mentioned that future studies should quantify the natural flower:leaf biomass proportion and evaluate which flower:leaf proportions often generate LMEs (Lines 545-548).

References:

Hector, A. (1998). The effect of diversity on productivity: detecting the role of species complementarity. Oikos, 82(3), 597-599.

Loreau, M. (1998). Separating sampling and other effects in biodiversity experiments. Oikos, 82(3), 600-602.

Loreau, M., & Hector, A. (2001). Partitioning selection and complementarity in biodiversity experiments. Nature, 412(6842), 72-76.

**• Line 26-27: 'while leaf litter had a higher concentration of Ca, Mg, Na' – What are the implications of this result? How is this important to understand the characteristics of leaf litter and their role on litter-mixing effects?**

**Response:** Several studies, especially in tropical environments, point to the limitation of decomposition by other nutrients than N and P. Thus, the combination litter with higher concentrations of N, P, and K (i.e. flower litter) with litter with higher content of Mg, Ca, and Na (i.e. leaf litter) favor the complementary effects. The information above is in the discussion (Lines 432-439). To reinforce the complementary effects between the litter types, we rephrased the sentence in the abstract after the sentence mentioned to reinforce the complementary effects between flower and leaf litter: "Our results demonstrate the importance of litter-mixing effects between flower and leaf litter via complementary effects" (Lines 25-26).

**• Consider mentioning the labile and recalcitrant nature of flower and leaf litter in the abstract.**

**Response:** We thank the reviewer for the comment. We mentioned the labile and recalcitrant nature of the two litter types used in the abstract: "Flower litter had a higher concentration of labile C-compounds, N, P, and K and lower lignin concentrations

representing a labile litter; while leaf litter had a higher concentration of lignin, Ca, Mg, and Na, representing a refractory litter " (Lines 23-26).

• **Line 27: Consider incorporating this statement "To our knowledge, …" earlier in the abstract. Be concise. (e.g., Line 18, 'for the first time')**

**Response:** We thank the reviewer for the comment. We replaced the sentence: "for the first time" with "To our knowledge" (Line 16). Then, we rephrased the sentence in line 25 as: "Our results demonstrate…" to avoid redundancy and repeating the same information twice.

• **Line 33: Include references across terrestrial and aquatic ecosystems • Line 48: add relevant reference(s)**

**Response:** We thank the reviewer for pointing this out, as suggested to the other reviewer, we included the relevant references (Lines 32-33):

Cebrian, J. and Lartigue, J.: Patterns of Herbivory and Decomposition in Aquatic and Terrestrial Ecosystems, Ecol. Monogr., 74, 237–259, 2004

Tiegs, S. D., Costello, D. M., Isken, M. W., Woodward, G., McIntyre, P. B., Gessner, M. O., et al.: Global patterns and drivers of ecosystem functioning in rivers and riparian zones, Sci. Adv., 5, eaav0486, https://doi.org/10.1126/sciadv.aav0486, 2019.

• **Line 89-105: Consider moving this background information earlier for better flow.**

**Response:** We thank the reviewer. We moved the information about the labile and refractory nature of flower and leaf litter after the paragraph about the possible occurrence of LMEs at the intra-specific level (Lines 71-86).

• **Consider moving section 2.4 earlier in the methods, as this section pertains to characterizing the initial chemical compositions of flower and leaf litter. I would prefer to read this before the experimental setup.**

**Response:** We thank the reviewer for the suggestion. We moved section 2.4, which turns into section 2.2, after only the section *Study site and species* (Lines 130-149).

• **Also, consider moving LMPexp (eq. 3) and RME (eq. 4) to section 2.4 and reserve section 2.5 for statistical analysis.**

**Response:** We thank the reviewer for the suggestion. We moved equations 3 and 4 to section 2.4- *Measurements of the litter mass remaining* (previously section 2.5; Lines 233-247).

• **Line 203-204: Briefly explain the analytical methods/protocols used to determine K, Ca, Mg, and Na contents**

**Response:** We thank the reviewer for the suggestion. We add more explanations about the determination of K, Ca, Mg, and Na contents: "Potassium (K), calcium (Ca), and manganese (Mg) were determined in flame atomic emission spectroscopy after nitro-perchloric digestion (Sarruge and Haag, 1974). Sodium (Na) content was estimated via flame atomic emission spectroscopy (Robertson et al., 1999)" (Lines 137-139).

• **Line 215: three 3 times à three times**

**Response:** We thank the reviewer for highlighting this mistake, but we deleted this sentence as the following suggestion from the other reviewer: "Lines 213 - 216: It is not clear why did you measure WHC and leaching along the gradient of variation of flower:leaf litter. Where are these results presented and discussed also?"

• **Line 222-225: Clarify this for better understanding.**

**Response:** We thank the reviewer for the comment. We rephrased the sentence as: "To quantify the LMEs for the whole mixture, we compared the observed ($LMR_{obs}$) and the expected ($LME_{exp}$) LMR (Loreau, 1998)" (LInes 233-234).

**• Line 278: consistent to à consistent with**

**Response:** We thank both reviewers for pointing out this grammar error. We changed the sentence as suggested: "This pattern was consistent with that observed in the aquatic experiment, and the values of leaf and flower litter decomposing in monoculture were significantly different from each other." (Lines 292-293).

**• Line 365: Specify what the 'second prediction' was.**

**Response:** We rephrased the sentence, specifying the second prediction: "Secondly, our results strongly supported that the interaction between flower and leaf would result in LMEs since the occurrence of LMEs to the flower and leaf litter mixture were consistent in terrestrial and aquatic environments" (Lines 383-384).

**• Line 440: Revisit this sentence "However, ……" for clarification.**

**Response:** We thank the reviewer. We rephrased the sentence to make clear the main factors explaining the antagonistic effects in treatment with lower flower litter biomass to: "Otherwise, the antagonistic effects observed in treatments with lower biomass of flower litter may be associated with the preferential feeding of decomposers on flower litter, and the low energy provided by the LOM was not enough to induce the degradation of the ROM (Cheng, 2009; Wang et al., 2015). It is important to note that we did not use labeled material to clearly distinguish the ROM and the LOM dynamic as classically done in priming experiments. Therefore, our priming related interpretation must be taken with due care" (Lines 459-463).

• **Line 479-491: Elaborate on the effects of soil invertebrates and their importance in litter-mixing effects in decomposition. Briefly mention how they affect LMEs. Include references for both caveats mentioned.**

**Response:** We thank the reviewer. We added a sentence indicating the main mechanisms related to soil fauna on LMEs including references to the gaps mentioned: "Secondly, the absence of macrofauna in our experiment could limit an accurate estimation of LMEs through flower and leaf litter interaction, since the presence of macro-fauna has been repeatedly shown to be an important factor in determining the occurrence and magnitude of synergistic LMEs on decomposition through litter fragmentation and decomposers complementary use of litter resources (Hättenschwiler and Gasser, 2005; Njoroge et al., 2022, 2023). Therefore, in future studies the inclusion of macro-fauna could be important to quantify the real impact of flower and leaf litter interaction in nutrient dynamics in ecosystems" (Lines 504-509).

• **Line 539: the importance litter from … à the importance of litter from …**

**Response:** We thank the reviewer. We changed the sentence as suggested (Lines 565).

• **Strengthen the conclusions by integrating key points made in the results and discussion sections. I recommend that the authors bolster the conclusions by reiterating important implications of the study, such as those outLined in Line 68-71 and Line 402-404.**

**Response:** We thank the reviewer for pointing this out. We strengthen the conclusions by adding, the importance of intra-specific variability of LMEs; how the interaction between flower and leaf litter could indicate the importance of functional dissimilarity rather than taxonomic diversity on LMEs; the effects above mentioned could be more relevant in low-diversity communities. Lastly, future studies could evaluate the generalities of our results to investigate the effects of flower and leaf litter interaction on LMEs in other species.

The new text excerpt is: "Our findings highlight the importance of litter from plant reproductive organs for LMEs in ecosystems, which could substantially contribute to changes in nutrient and carbon dynamics. Our results highlight the importance of intra-specific variability among organs indicating the occurrence of LMEs could be more dependent on litter dissimilarity than taxonomic richness, suggesting the potential relevance of LMEs at intra-specific levels in low-diversity communities. Although recent studies have evidenced the role of reproductive organs in increasing the decomposition of organic matter in the natural environment in both terrestrial (de Paz et al., 2018; Schmitt and Perfecto, 2020) and aquatic (Rezende et al., 2017) ecosystems, it is necessary to evaluate the general patterns and the possible mechanisms driving the effects of litter from reproductive organs on LMEs in each environment. We found evidence of the central role of complementarity mechanisms in the occurrence, magnitude, and direction of LMEs. In the terrestrial environment, the LMEs varied as a function of flower-to-leaf litter biomass proportion in the litter mixture, indicating the potential importance of interplay between resource quality and quantity in determining niche partitioning among microbial decomposers. To understand the generalities of the second role of flowers on litter decomposition, it is important to verify the generality of our results found for *T. aurea*. Future studies should investigate the generalities of flower and leaf litter on LMEs at intra- and inter-specific levels, as well as the potential role of flower litter in affecting direct and indirect mechanisms of LMEs on litter decomposition across a large variety of plant species, an aspect that has been completely neglected in the literature about the effects of litter mixing on decomposition" (Lines 565-579).